# Allosteric control of dynamin-related protein 1 through a disordered C-terminal Short Linear Motif

Isabel Pérez-Jover [1,2,9], Kristy Rochon [3,9], Di Hu [4,9], Mukesh Mahajan [4], Pooja Madan Mohan[4], Isaac Santos-Pérez[5], Julene Ormaetxea Gisasola[1,2], Juan Manuel Martinez Galvez[1,2], Jon Agirre [6], Xin Qi [4,7], Jason A. Mears[3,7,8], Anna V. Shnyrova [1,2] ✉ & Rajesh Ramachandran [4,8] ✉

The mechanochemical GTPase dynamin-related protein 1 (Drp1) catalyzes mitochondrial and peroxisomal fission, but the regulatory mechanisms remain ambiguous. Here we find that a conserved, intrinsically disordered, six-residue Short Linear Motif at the extreme Drp1 C-terminus, named CT-SLiM, constitutes a critical allosteric site that controls Drp1 structure and function in vitro and in vivo. Extension of the CT-SLiM by non-native residues, or its interaction with the protein partner GIPC-1, constrains Drp1 subunit conformational dynamics, alters self-assembly properties, and limits cooperative GTP hydrolysis, surprisingly leading to the fission of model membranes in vitro. In vivo, the involvement of the native CT-SLiM is critical for productive mitochondrial and peroxisomal fission, as both deletion and non-native extension of the CT-SLiM severely impair their progression. Thus, contrary to prevailing models, Drp1-catalyzed membrane fission relies on allosteric communication mediated by the CT-SLiM, deceleration of GTPase activity, and coupled changes in subunit architecture and assembly-disassembly dynamics.

Intrinsically disordered proteins (IDPs) and structured proteins that contain intrinsically disordered regions (IDRs) are ubiquitous comprising nearly half of the human proteome[1–3]. In contrast to the relatively stationary loops and turns that connect secondary structure elements in compactly folded protein domains, IDRs persist as a highly dynamic conformational ensemble, which in many instances undergoes a localized disorder-to-order structural transition upon partner interactions (with protein, lipid, or nucleotide) and/or via various post-translational modifications (PTMs)[4–6]. Consequently, IDRs function as regulatory nodes or hubs that govern host protein function by transcribing biological information from multiple interactions and modifications into discernible alterations in local protein fold, dynamics, and macromolecular assembly, including protein condensation via liquid-liquid phase separation (LLPS)[7–10].

This generalized description of IDR structure and function also pertains to dynamin-related protein 1 (Drp1), a self-assembling, multidomain GTPase that mechanochemically constricts tubular membrane intermediates en route to mitochondrial fission[11,12]. Drp1 contains

[1]Department of Biochemistry and Molecular Biology, University of the Basque Country, 48940 Leioa, Spain. [2]Instituto Biofisika, CSIC, UPV/EHU, 48940 Leioa, Spain. [3]Department of Pharmacology, Case Western Reserve University School of Medicine, Cleveland, OH 44106, USA. [4]Department of Physiology and Biophysics, Case Western Reserve University School of Medicine, Cleveland, OH 44106, USA. [5]Electron Microscopy and Crystallography Center for Cooperative Research in Biosciences (CIC bioGUNE), Bizkaia Science and Technology, Park Bld 800, 48160-Derio Bizkaia, Spain. [6]York Structural Biology Laboratory, Department of Chemistry, University of York, Heslington, YO10 5DD York, UK. [7]Center for Mitochondrial Diseases, Case Western Reserve University School of Medicine, Cleveland, OH 44106, USA. [8]Cleveland Center for Membrane and Structural Biology, Case Western Reserve University School of Medicine, Cleveland, OH 44106, USA. [9]These authors contributed equally: Isabel Pérez-Jover, Kristy Rochon, Di Hu. ✉e-mail: anna.shnyrova@ehu.eus; rxr275@case.edu

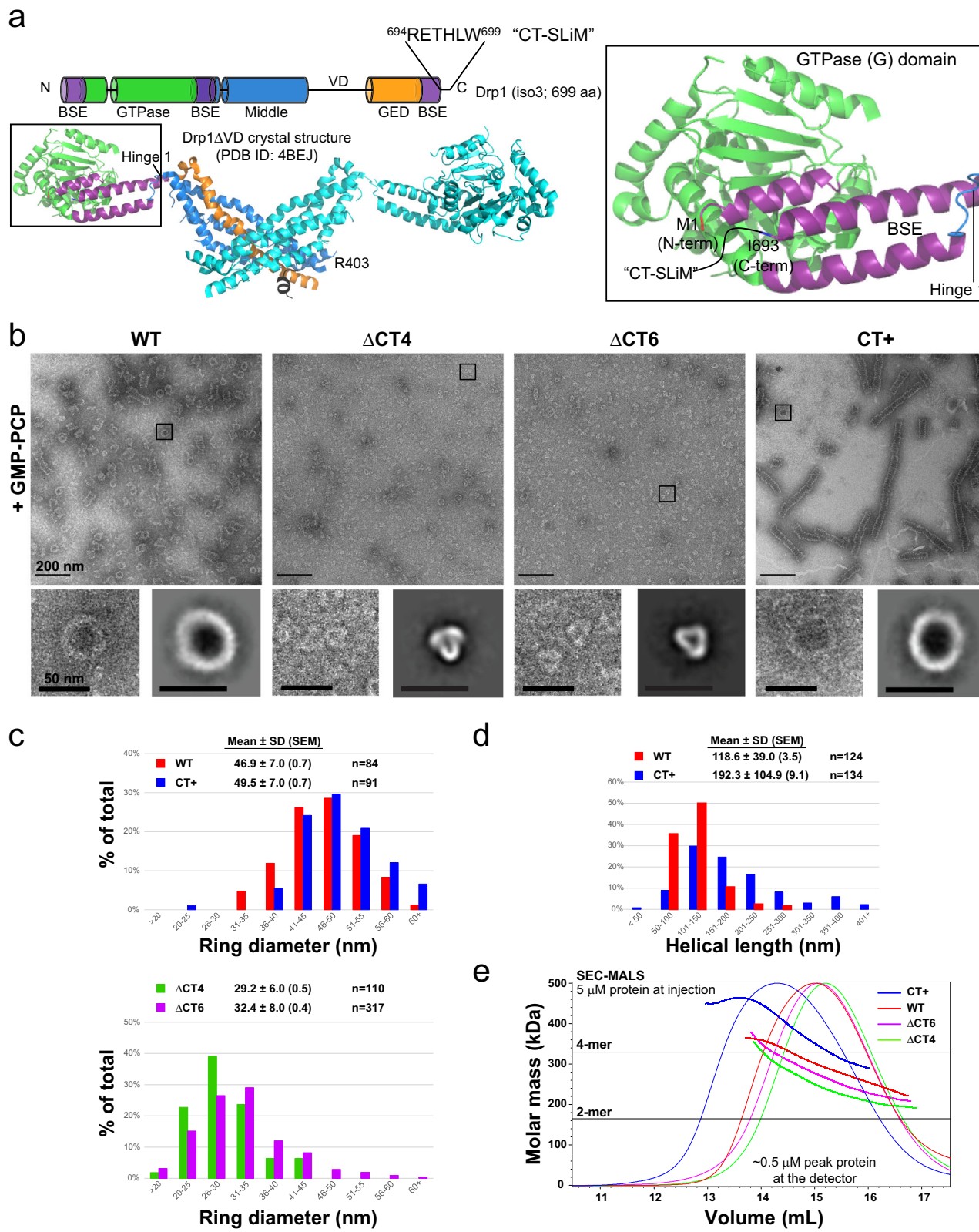

multiple IDRs, ranging up to 134 amino acid (aa) residues in length, which make up >20% of its overall sequence[11,13,14] (Supplementary Fig. 1a). Yet, many of these IDRs are either absent or unresolved in any available Drp1 X-ray[13] or cryo-EM structure[15,16] to date (Fig. 1a and Supplementary Fig. 1b), obscuring further characterization. These include: (i) Molecular Recognition Features (MoRFs)[8,10] of ~10−25 aa residues, such as MoRFs-1 and −2 embedded within the

largely disordered variable domain (VD), enabling direct Drp1-membrane interactions[17] and (ii) Short Linear Motifs (SLiMs)[8,10] of ~3−12 aa residues nested within highly structured domains, such as the G-domain '80-loop' and stalk 'L1N' loop that direct protein-protein interactions during Drp1 self-assembly[14]. One such IDR is a unique stretch of ~6 aa residues at the Drp1 extreme C-terminus, which we call the CT-SLiM (Fig. 1a and Supplementary Fig. 1a, b) that, unlike other

**Fig. 1 | CT-SLiM modifications affect Drp1 oligomerization propensity and helical geometry. a** The location and polypeptide sequence of the CT-SLiM in Drp1 isoform 3 primary structure is shown. The crystal structure beneath corresponds to the Drp1ΔVD dimer with a color-coded representation of domain arrangement in a monomer. BSE (purple) is the bundle signaling element. The stalk comprises a four-helical bundle composed of discontinuous middle (blue) and GED (GTPase effector domain; orange) regions. GTPase (G) domain is shown in green. Connecting black lines represent a few prominent IDRs in Drp1. The VD connects the middle and GED regions, whereas the 80-loop and LIN loops are nested within the G and stalk (middle) domains, respectively. The inset is a zoomed-in view of the BSE showing the well-resolved Drp1 N-terminal BSE helix (beginning from aa residue 1). The last six residues of the Drp1 C-terminus (R694-W699), an IDR which we call the CT-SLiM and represented here by a curved black line, remain disordered. I693, the last resolved residue of the C-terminal BSE helix is highlighted. **b** Representative NS-EM images of WT Drp1 and CT variants in the presence of the non-hydrolyzable GTP analogue, GMP-PCP. Scale bar, 200 nm. The left inset under each panel shows a zoomed-in view of the boxed region in the above micrograph, whereas 2D class averages of the predominant oligomer (ring) morphology are shown to their right. Insets scale bar, 50 nm. Data shown here are for mouse CT+ Drp1. Human CT+ Drp1 data are shown in Supplementary Fig. 2a. Histograms showing the distribution of assessed ring diameter (**c**) and helical polymer length (**d**) for WT Drp1 and CT variants. ΔCT4/6 Drp1 do not form helical polymers. Mean ± SD (SEM) is indicated. n is the number of particles. **e** SEC-MALS elution and molar mass profiles of human WT Drp1 and CT variants sieved through a Superose 6 10/300 GL column. When injected at 5 μM, Drp1 is diluted to ~0.5 μM peak concentration on arrival at the LS and dRI detectors. Horizontal lines indicate the theoretical masses of a Drp1 dimer (2-mer) and tetramer (4-mer).

IDRs in Drp1, is highly conserved among metazoans (Supplementary Fig. 1a). However, its function(s) remain largely unexplored.

Recent studies have indicated that this CT-SLiM constitutes an atypical PDZ domain binding motif (PBM) that specifically interacts with the PDZ domain-containing adaptor protein GIPC-1 (GAIP interacting protein, C-terminus 1)[18,19]. CT-SLiM-bound GIPC-1, in turn, associates with the F-actin minus-end-directed motor myosin VI (MYO6) to guide Drp1 presumably to F-actin-pre-constricted mitochondrial division sites[18–21]. However, whether or how direct GIPC-1-Drp1 interactions via the CT-SLiM influence Drp1 structure and/or function remains unknown.

Here, using a comprehensive toolkit of structural, cell biological, and in vitro reconstitution approaches, we show that a deletion (ΔCT) or a non-native extension (CT + ) of the CT-SLiM distinctly alters Drp1 conformational dynamics, oligomerization propensity, self-assembly geometry, and cooperative GTPase activity, in addition to differentially affecting Drp1 capacity to remodel target membranes. We demonstrate that whereas the ΔCT variants exhibit a predictable loss-of-function by either altering or inhibiting membrane fission both in vitro and in vivo, the CT+ variants display an aberrant gain-of-function by robustly catalyzing membrane fission in vitro, while remaining repressed in mediating mitochondrial fission in vivo. By contrast, WT Drp1, which is limited to constricting membranes on its own in vitro, remarkably progresses toward membrane fission upon native CT-SLiM-effected GIPC-1 interactions. Taken together, our data indicate a critical role for the native CT-SLiM in governing Drp1 structure, conformational dynamics, and mechanoenzymatic membrane remodeling activity. Furthermore, key protein partner interactions of Drp1, such as that of the CT-SLiM, emerge as an essential regulatory element in the allosteric control of Drp1 function during mitochondrial fission.

## Results

### CT-SLiM modifications alter Drp1 self-assembly propensity and geometry

To discern the role of the CT-SLiM, we generated a host of Drp1 variants with either truncated or extended C-termini (Supplementary Fig. 1c). The truncated variants had the last four (ΔCT4) or six (ΔCT6) residues of the native CT-SLiM removed in order to separate the potential electrostatic ($R^{694}E^{695}$) and hydrophobic ($L^{698}W^{699}$) contributions of this segment to Drp1-partner protein interactions[18]. Conversely, the extended variants (CT + ) had non-native sequences of different lengths and composition, including affinity and epitope tags, appended to the CT-SLiM. These non-native extensions were introduced to isolate the backbone carboxylate moiety of the C-terminal residue, a requirement for high-affinity PDZ domain binding[22], from the predicted PDZ domain recognition sequence in Drp1 ($^{696}THLW^{699}$)[18]. In addition, we sought to determine the influence of non-native C-terminal extensions on Drp1 structure and function, which in past studies have produced confounding and conflicting results[23–25]. To enable purification, WT Drp1 and select CT variants were modified at the N-terminus with a $His_6$ affinity tag (see Methods), which as previously shown[26,27] did not affect Drp1 self-assembly or GTPase activity in vitro. Besides, N-terminally epitope (Myc)-tagged Drp1 effectively restored mitochondrial fission in Drp1 knockout (KO) cells[17,26] indicating that these N-terminal modifications neither affect Drp1 function in vivo.

Negative-stain electron microscopy (NS-EM) analysis revealed considerable alterations in the Drp1 oligomer structure due to the CT-SLiM modifications. In the presence of the non-hydrolyzable GTP analogue, GMP-PCP, which mimics GTP binding and promotes Drp1 helical self-assembly in solution[26], WT Drp1 characteristically formed a mixture of oligomeric rings and higher-order spirals of a consistent diameter and length (Fig. 1b–d). In contrast, the ΔCT4 and ΔCT6 variants failed to assemble into any such regular higher-order structures. Instead, the ΔCT4 and ΔCT6 variants predominantly constituted triangularly shaped nubs of much smaller dimensions with little to no indication of further higher-order self-assembly (Fig. 1b, c). Conversely, the CT+ variant formed consistently longer supramolecular helical assemblies, although similar in overall helical diameter to WT (Fig. 1b–d).

Size-exclusion chromatography-coupled multi-angle light scattering (SEC-MALS) analyses of these variants in the nucleotide-free apo state at physiologically relevant concentrations in solution[28] (~0.5 μM at peak detection upon ~10-fold SEC dilution) revealed further differences in their oligomerization properties relative to WT (Fig. 1e). The ΔCT4 and ΔCT6 variants exhibited a sharp dimer-tetramer equilibrium similar to WT, albeit tending marginally toward minimal dimers under the conditions. In contrast, the extended CT+ variant largely favored higher-order oligomers consistent with its enhanced helical self-assembly in the presence of GMP-PCP (Fig. 1b, Supplementary Fig. 2a, b). This greater oligomerization propensity of the CT+ variant relative to WT was evident over a wide range of protein concentrations (Supplementary Fig. 2c). Besides, it was independent of the non-native CT+ sequence as this tendency was also manifest in a CT+* variant containing an extension of a different length (14 aa residues) and composition (Supplementary Fig. 2d). Shortening the non-native CT sequence of the CT+ variant from 24 to 9 aa residues by proteolytic cleavage (CT+$^{sh}$) reduced its higher-order oligomerization propensity (Supplementary Fig. 2e), although this remained noticeably greater than that of WT Drp1. On the other hand, shortening the N-terminal tag sequence from 36 to 7 aa residues had no palpable effect on Drp1 oligomerization (Supplementary Fig. 2f).

These data indicated that the disordered Drp1 CT-SLiM is a critical determinant of Drp1 self-assembly and helical propagation.

### CT-SLiM modifications alter Drp1 conformational dynamics and structure

To gain insight into the molecular mechanisms underlying CT-SLiM function, we used NS-EM and performed 2D image classification to assess the impact of the various CT modifications on Drp1 subunit

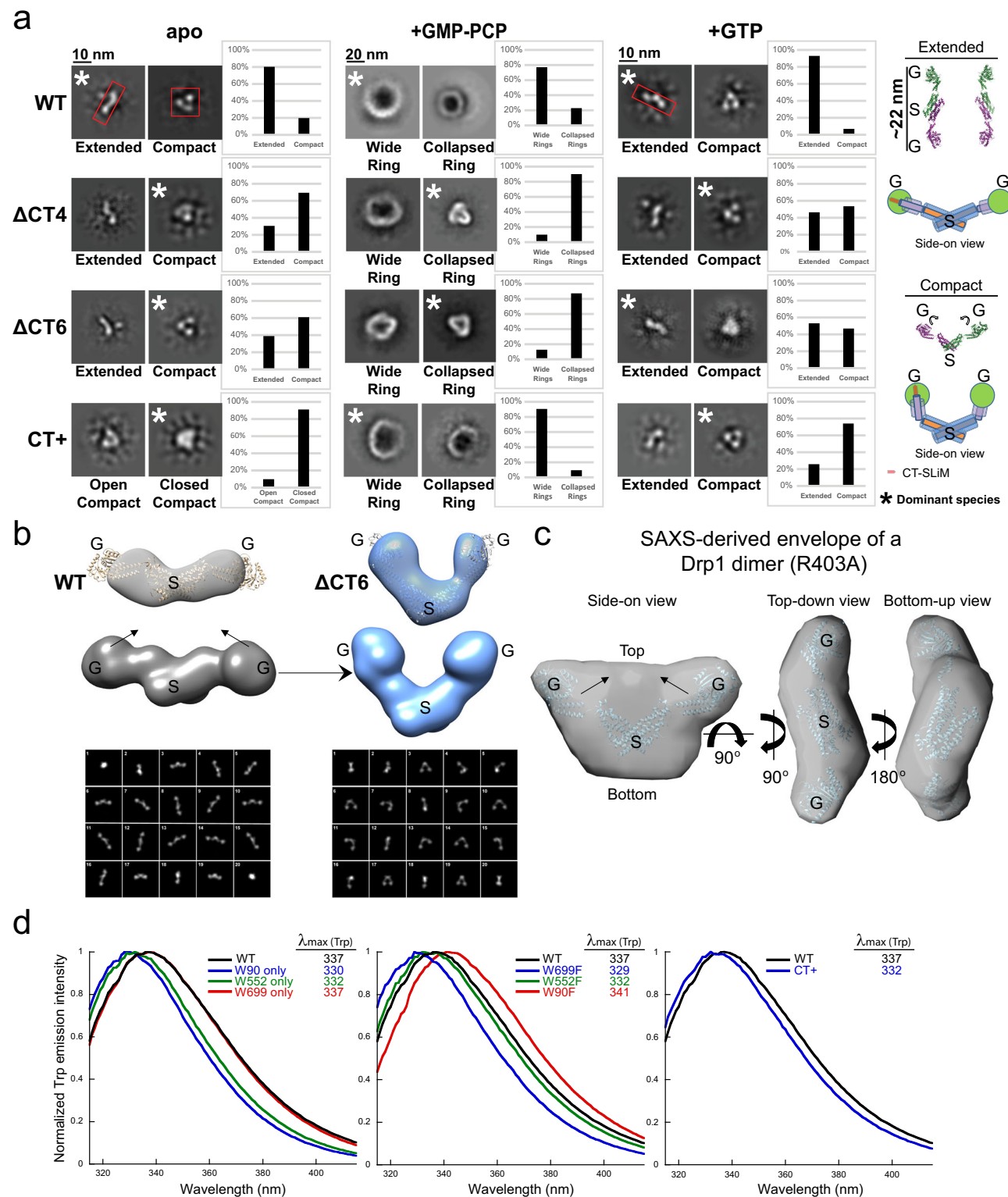

conformational dynamics and self-assembly under different nucleotide-bound states (Fig. 2a, Supplementary Fig. 3a, b).

In the apo state, we detected two different orientations for the WT Drp1 dimer−an S-shaped top-down (or bottom-up) orientation and a V-shaped side-on orientation with prominent densities evident for the dimeric stalk and the two individual GTPase (G) domains (Fig. 2a, Supplementary Fig. 3b, c). Notably, in the S-shaped orientation, the G domains were set far apart, whereas in the V-shaped orientation, the G domains appeared to be positioned in close

proximity. Remarkably, the S-shaped structure was never found for the CT+ variant in the apo state and was relatively poorly sampled by the ΔCT4/6 variants (Fig. 2a, Supplementary Fig. 3b, c). As differential grid deposition or preferred orientations were unlikely to be influential factors owing to the minimal nature of the CT modifications involved (CT-SLiM deletion or a short non-native extension), we reasoned that these two orientations likely correspond to two different solution conformations of the Drp1 dimer that interconvert dynamically (Fig. 2a, Supplementary Fig. 3c).

**Fig. 2 | CT-SLiM modifications alter Drp1 conformational dynamics. a** NS-EM 2D class averages of dimers and oligomeric rings in the apo, GMP-CPC-bound, and GTP hydrolysis states for human WT Drp1 and CT variants. Drp1 dimer conformation in the apo and GTP hydrolysis states is classified as either extended or compact, with the latter classified further into open-compact or closed-compact states. Oligomeric ring morphology in the presence of GMP-CPC is classified into wide and collapsed ring states. Top-down and side-on views of the ΔVD Drp1 dimer crystal structure (PDB ID: 4BEJ) as well as color-coded cartoon illustrations of domain rearrangements between the extended and compact states are shown at the far-right corner. G refers to the G domain, whereas S refers to the stalk. The number of particles in the extended (E) and compact (C) conformations in the apo and +GTP states, and displaying Wide Ring (WR) and Collapsed Ring (CR) morphologies in the +GMP-CPC state are shown in Methods. **b** i) WT and ΔCT6 Drp1 3D densities rendered from cryoSPARC homogeneous refinement processing, ii) WT and ΔCT6 Drp1 idealized models generated from the ΔVD Drp1 dimer crystal structure (PDB ID: 4BEJ) by thresholding the structure to 40 Å for WT and by repositioning the G domains and thresholding to 40 Å for ΔCT6 Drp1, and iii) 2D class averages generated from the WT and ΔCT6 Drp1 idealized models are shown in rows. **c** Views of the SAXS-derived ab initio envelope of dimeric R403A Drp1 overlaid with the ΔVD Drp1 dimer crystal structure (PDB ID: 4BEJ). **d** Normalized Trp emission spectra of human WT Drp1 in comparison to single W-only mutants, single W-to-F mutants, and the CT+ Drp1 variant are shown, respectively, from left to right. The wavelength of maximum emission ($\lambda_{max}$) is indicated.

For the apo WT Drp1 dimer, the extended S-shaped conformer was detected at a ~4-fold greater incidence than the compact V-shaped conformer, indicating a greater residence time for the native dimer in the extended conformation (Fig. 2a). These data indicated that the native CT-SLiM restricts Drp1 conformational dynamics in solution and retains Drp1 predominantly in the extended conformation, whereas its absence or non-native extension in the ΔCT4/6 and CT+ variants, respectively, differentially relieves this auto-inhibition favoring their conversion, to varying extents, to the alternate compact conformation. Moreover, in the presence of GMP-CPC, the oligomeric rings formed by the ΔCT4/6 and CT+ variants were largely irregular or poorly ordered (Fig. 2a) suggesting that the native CT-SLiM also functions as a spacer that sets the register and geometry of inter-subunit interactions during nucleotide-dependent helical self-assembly. Furthermore, unlike WT, which reverted to the extended dimer conformation upon GTP hydrolysis, the CT+ and ΔCT4/6 variants largely remained in the compact conformation (Fig. 2a, Supplementary Fig. 3d). These data indicated that the compact CT+ Drp1 conformer likely mimics an assembly-primed state based on its greater higher-order oligomerization propensity relative to WT both in absence and presence of nucleotide.

3D reconstruction from 2D class averages of the extended WT and compact ΔCT6 Drp1 conformations further allowed us to dock the available crystal structure of the ΔVD Drp1 dimer and examine the nature of the conformational rearrangements (Fig. 2b, Supplementary Fig. 3c). With the extended conformation, the G domains of the docked ΔVD Drp1 dimer stretched beyond the computed edge densities. These data suggested that either the ΔVD variant is in an alternate conformation compared to WT[14], with the WT G domains tucked in toward the stalk as recently indicated[16], or that our reconstructed structure remains partially unresolved owing to inherent dynamics around hinge 1 at the BSE-stalk intersection (Fig. 1a). Fitting of the ΔVD Drp1 dimer structure into the computed 3D volume of the compact conformation, however, required a large-scale repositioning of the G domains around hinge 1 (Fig. 2b). Modeling of these two conformations using the crystal structure (Fig. 2b) and back projection of 2D class averages from the computed 3D volumes (Fig. 2b) revealed that the compact conformation sampled by ΔCT6 Drp1 is not observed in any of the projected 2D class averages for the extended WT Drp1 dimer. These data indicated that the observed extended and compact Drp1 forms are indeed conformationally distinct.

To confirm the large-scale flexibility of the Drp1 dimer as indicated by the EM data, we mapped the conformational landscape of a minimal Drp1 dimer in solution using small-angle X-ray scattering (SAXS)[29] as an orthogonal approach (Fig. 2c and Supplementary Fig. 4). The heterogenous mix of dimers, tetramers, and higher-order oligomers present in dynamic equilibrium for WT Drp1 and the CT variants is incompatible with SAXS and cannot be analyzed. Therefore, we employed a R403A mutation in Drp1 (R399A in Dyn1[30]) (Fig. 1a) that restricts Drp1 predominantly to a minimal dimer in solution (Supplementary Fig. 4a), similar to the minimal Dyn1 dimer previously assessed by SAXS[31]. Remarkably, ab initio reconstruction of the most probable low-

resolution molecular envelope for the R403A Drp1 dimer revealed an overall shape that was compatible with both the extended and compact conformations (Fig. 2c), with sufficient volume present between the G domains and below the stalk of the overlaid ΔVD Drp1 dimer crystal structure to accommodate both shapes. Thus, the minimal Drp1 dimer in solution is highly dynamic and capable of interconversion between extended and compact states.

To understand the molecular basis of the CT+ Drp1 variant's distinctively compact conformation and gain-of-function in self-assembly, we used AlphaFold[32] to predict the influence of the CT+ sequence extension on Drp1 structure. Remarkably, the computational data suggested that whereas the N-terminal His$_6$ affinity tag in our WT Drp1 was mostly disordered, the non-native CT extension in CT+ Drp1 propagated as a α-helix in close apposition to the top of the G domain, potentially constraining dynamics at the adjacent nucleotide-sensitive G domain-BSE interface (Supplementary Fig. 5a). Consistent with this, a direct comparison of the top-ranked structures in isolation (Supplementary Fig. 5b) and upon superposition into the available Drp1 polymer cryo-EM structure (Supplementary Fig. 5c), revealed a slight inward buckling of the G domain toward the BSE in CT+ Drp1 compared to WT Drp1. In addition, given the proximity of the CT-SLiM to the stalk of the adjacent monomer in the Drp1 polymer (Supplementary Fig. 5c), the modeling data further indicated that the CT+ extension may influence Drp1 subunit-subunit interactions during higher-order helical self-assembly.

We used intrinsic Tryptophan (Trp) Fluorescence Spectroscopy (iTFS)[33,34] to experimentally validate these in silico predictions (Fig. 2d). Trp emission is highly sensitive to its microenvironment and therefore serves as an accurate probe of protein conformation or conformational changes[25,26]. When excited selectively at $\lambda = 295$ nm, the Trp emission spectrum is blue-shifted (peaking at shorter wavelengths) when present in a nonpolar environment, and red-shifted (peaking at longer wavelengths) when exposed to a polar or aqueous milieu. Drp1 contains three native Trp residues at positions 90, 552 and 699 (ubiquitous isoform 3 numbering; Fig. 1a). Of these, only W90 present in the G domain is structurally resolved[13] (Supplementary Fig. 5d), whereas W552 and W699 are located in the disordered VD and CT-SLiM, respectively (Supplementary Fig. 1a). Using site-directed Drp1 mutants that retained only one of the three native Trp or that contained only a single native Trp-to-Phe substitution, we ascertained that Drp1 Trp emission primarily originates from W699, the terminal residue of the CT-SLiM.

Consistent with the partial burial of W90 in the Drp1 G domain structure[13] (Supplementary Fig. 5d), the W90-only mutant displayed a pronounced blue shift in Trp emission relative to WT (Fig. 2d). Similarly, the W552-only mutant also exhibited a significant blue shift, albeit less than that of the W90-only mutant, indicating that W552 is also partially occluded from solvent in the VD conformational ensemble (Fig. 2d). By contrast, the W699-only mutant was pronouncedly red-shifted and was identical to WT in emission spectra (Fig. 2d). These data indicated that W699 in WT Drp1 is solvent accessible, and is the primary emitter largely owing to its location within Trp-Trp homo-

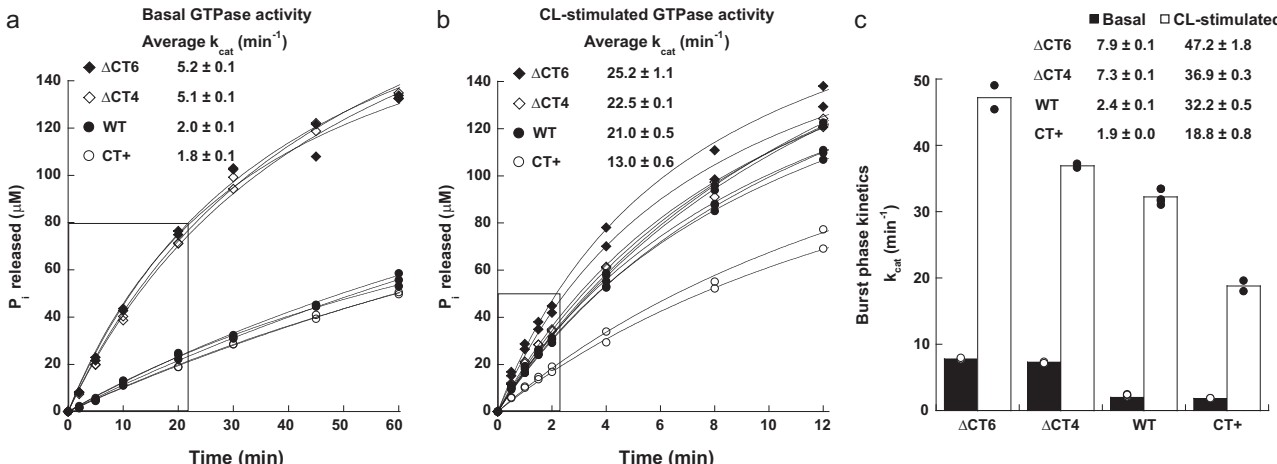

**Fig. 3 | CT-SLiM modifications differentially affect Drp1 GTPase activity.** Relative basal (**a**) and CL-stimulated GTPase activities (**b**) of WT Drp1 and CT variants at 0.5 μM each in the absence and presence of CL-containing liposomes (150 μM total lipid). GTPase activities of the CT variants were measured in parallel in comparison to WT from two independent protein alone or protein-lipid mixture samples (n = 2 measurements for each variant under either condition; n = 3 and 5 for WT basal and CL-stimulated activities, respectively). The concentration of inorganic phosphate ($P_i$) released is plotted against time. Individual data points and best fit traces for each time course are shown. In **a** and **b**, the average turnover number ($k_{cat}$) ± SEM derived from a linear regression analysis of both pre-steady state (burst phase; boxed regions) and steady-state pre-steady-state kinetics data points is indicated above. **c** $k_{cat}$ for the burst-phase pre-steady-state kinetics only from each measurement. Individual data points are overlaid on bar plots representing the average. Burst-phase $k_{cat}$ ± SEM is indicated above.

FRET distance (~24 Å)[33] of the high-energy FRET donor, W90 (Supplementary Fig. 5d). Consistent with this assessment, the W90F mutant was red-shifted (by 11 nm) compared to the W90-only mutant, whereas the W699F mutant was blue-shifted (by 8 nm) relative to the W699-only mutant (Fig. 2d). The W552F mutant, on the other hand, did not experience any such change (Fig. 2d). These data confirmed that W699 in the WT Drp1 CT-SLiM is exposed to water and is highly responsive to its local environment. Notably, by contrast to WT Drp1, CT+ Drp1 emission was blue-shifted (by 5 nm) indicating that W699 in CT+ Drp1 is instead buried and relatively solvent inaccessible (Fig. 2d). No such change in Trp emission was observed when the 36 aa-residue N-terminal His$_6$ tag of WT Drp1 was replaced by a relatively short 7-aa residue overhang (Supplementary Fig. 5e), indicating that the difference in the environment of CT-SLiM in CT+ Drp1 is primarily due to the non-native CT extension. Thus, together with the cryo-EM data and AlphaFold predictions, the iTFS data demonstrated that non-native CT extension of CT+ Drp1 alters CT-SLiM microenvironment and overall Drp1 conformation. WT and CT+ Drp1 thus populate distinct conformational states.

## CT-SLiM modifications differentially affect Drp1 cooperative GTP hydrolysis

We next determined the impact of the various CT-SLiM modifications on Drp1 GTPase activity under basal conditions in solution and upon helical self-assembly on CL-containing membranes. In the apo state, the CT variants retained the characteristic capacity of WT Drp1 to self-assemble on, and tubulate, large CL-containing liposomes to narrow diameters (Supplementary Fig. 6a, b). Similarly, the CT variants also assembled on highly curved and preformed galactosylceramide-laden CL-containing lipid nanotubes (GalCer CL-NTs) identically to WT (Supplementary Fig. 6c). These data indicated that the various CT modifications do not adversely affect stalk-mediated Drp1 self-assembly on membranes.

Surprisingly, however, the ΔCT4 and ΔCT6 variants both exhibited a ~3-fold greater rate of GTP hydrolysis in solution compared to the CT+ variant and WT, which were similar in basal GTPase activity (Fig. 3a). By contrast, the CT+ variant displayed a ~2-fold lower activity compared to the ΔCT4/6 variants and WT when assayed on CL-containing liposomes (Fig. 3b). Analysis of the pre-steady state 'burst'

kinetics revealed that the ΔCT4/6 variants hydrolyzed GTP at a significantly faster rate than WT under both conditions (Fig. 3c). Similar trends also held up for the CT variants on GalCer CL-NTs on which ΔCT4/6 Drp1 hydrolyzed GTP at a significantly faster rate than both WT and CT+ Drp1 in the order: ΔCT6 > ΔCT4 > WT > CT+ (Supplementary Fig. 6d, e). Co-sedimentation analysis of the Drp1 variants on GalCer CL-NTs, in the absence and presence of GTP, demonstrated a greater steady-state association of the faster GTP-hydrolyzing ΔCT4/6 variants with the lipid templates than WT or CT+ Drp1 (Supplementary Fig. 6f). Consistent with this, NS-EM on GalCer CL-NTs in the presence of GTP revealed persistent self-assembly of the ΔCT6 variant on the lipid templates. Such phenotype was absent for both WT and CT+ Drp1, which showed widespread disassembly and membrane dissociation with GTP (Supplementary Fig. 7). Interestingly, in the presence of GMP-PCP, the ΔCT6 variant, relative to WT and CT+ Drp1, formed highly processive helical polymers, which in many instances extended beyond the lipid template (Supplementary Fig. 7).

Collectively, these data indicated that in the absence of the CT-SLiM, transition state-dependent inter-subunit G-domain dimerization, cooperative GTP hydrolysis, GDP/Pi release, and G-domain dimer disassembly critical for progressive rounds of GTP binding and hydrolysis are all significantly accelerated for the ΔCT4/6 variants, manifested in faster recycling and greater steady-state association with membranes. Conversely, for the same reasons, in the presence of a non-native CT extension that non-physiologically stabilizes inter-subunit Drp1 interactions and exaggeratedly promotes helical self-assembly in CT+ Drp1, GTP turnover and recycling on membranes appears to be decreased. Thus, faster dynamics in the absence of the CT-SLiM, and altered, slower dynamics in the presence of a non-native CT extension distinctively affect Drp1 cooperative GTPase activity relative to WT.

Together with its impact on Drp1 dimer structure, these data therefore raised the intriguing prospect that the native CT-SLiM functions as a 'kinetic timer' of Drp1's GTP hydrolysis rate and coupled membrane remodeling activity.

## CT-SLiM controls Drp1-catalyzed membrane fission in vitro

We therefore addressed whether the differential GTPase activity, and altered conformational and self-assembly dynamics of the CT variants relative to WT translated to distinct membrane remodeling

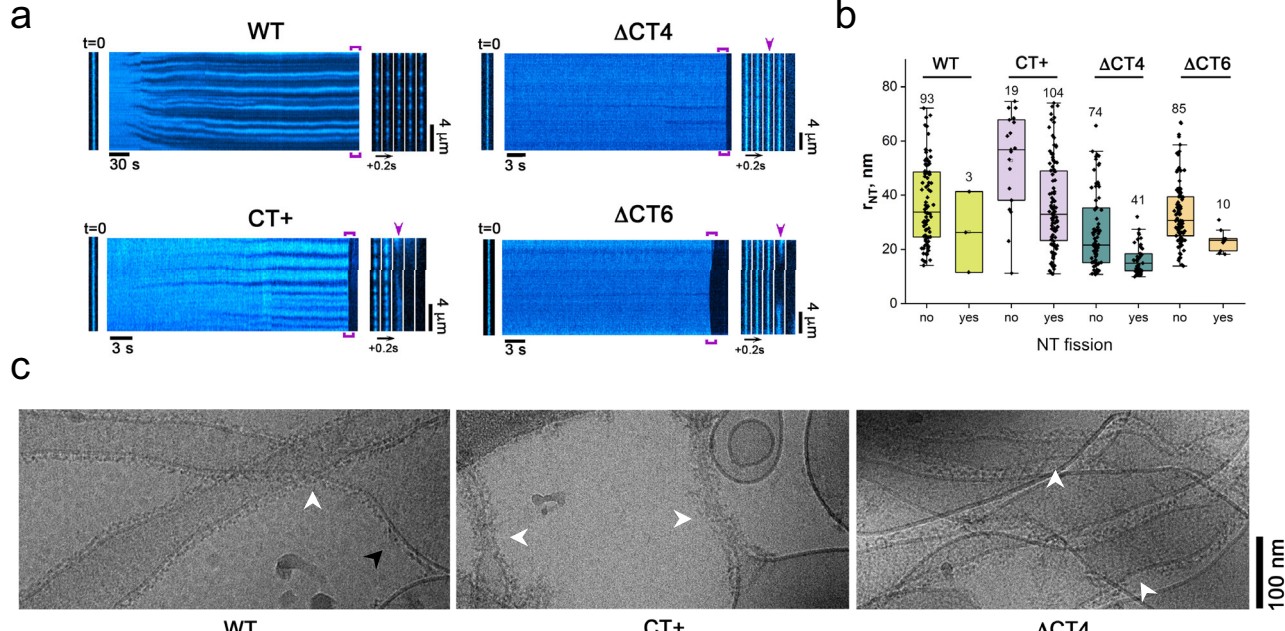

**Fig. 4 | CT-SLiM modifications differentially affect membrane remodeling and fission. a** Representative kymographs showing the remodeling of freely suspended NTs upon addition of 0.5 μM of WT Drp1 or CT variants in the presence of 1 mM GTP. NT membrane fluorescence is displayed in cyan pseudocolor for clarity. Left images correspond to the initial frame of the kymographs. Right image sequences correspond to the framed region of the kymographs. Purple arrows indicate NT fission. **b** Distributions of the radii of free-standing NTs that underwent fission (yes) or were only constricted (no) upon addition of 0.5 μM WT Drp1 or CT variants in the presence of 1 mM GTP. The numbers on top of each box represent the total number of NTs for each condition. Error bars are SD, $n = 3$ independent experiments. Box plots indicate median (middle line), 25th and 75th percentile (box) and outliers (whiskers). **c** Cryo-EM images showing WT Drp1, CT+ Drp1, and ΔCT4 Drp1 assembled on preformed NTs in the presence of 1 mM GTP. White arrowheads indicate Drp1 scaffolds on highly curved NT membranes. Black arrowhead shows curvature-adaptable assembly of WT Drp1 also on relatively flat (low curvature) membrane regions, not observed with CT+ Drp1.

phenotypes. To this end, we tested the efficacy of our CT variants in directing the scission of suspended lipid nanotubes (NTs) mimicking the mitochondrial outer membrane at pre-constricted mitochondrial division sites. NTs ranging from tens to hundreds of nanometers in diameter were formed between polymer micropillars in a microfluidic chamber[35] (see Methods). WT Drp1 and CT variants at 0.5 μM final concentration, corresponding to the estimated cytosolic concentration of Drp1[28], as well as the concentration at which the catalytic activity ($k_{cat}$) of WT Drp1 nears saturation[26], were then infused into the chamber in the presence of 1 mM GTP, while NT constriction and/or scission was monitored in real-time by fluorescence microscopy.

As previously shown[17,25], WT Drp1 did not effectively catalyze NT fission on its own (Fig. 4a, b). Instead, WT scaffold assembly on the NT resulted in NT constriction to a stable final radius of 14 ± 2 nm (measured at the membrane midplane) independent of the initial NT radii (Fig. 4a, Supplementary Movie 1). Surprisingly, both ΔCT4 and ΔCT6 Drp1 selectively mediated the fission of NTs with the highest initial curvatures (i.e., with radii < ~30 nm), with the ΔCT4 variant exhibiting the greater fission efficiency of the two (Fig. 4a, b, Supplementary Movies 2, 3). For both variants, the area of membrane constriction prior to fission appeared to be highly limited and narrow, being barely detectable by fluorescence microscopy (Fig. 4a). Thus, a partial or complete deletion of the CT-SLiM limits Drp1 scaffolding on membranes, probably due to the greater GTP hydrolysis rate of the ΔCT4/6 variants (Fig. 3 and Supplementary Fig. 6d, e), causing rapid oligomer disassembly and recycling on membranes (Supplementary Fig. 7).

In stark contrast to the ΔCT4/6 deletion variants, the CT+ extension variants elicited a robust constriction and fission of a broad range of initial NT curvatures (Fig. 4a, b, Supplementary Fig. 8a, Supplementary Movie 4). Notably, the scission efficiencies of these variants directly corresponded to their higher-order oligomerization propensities with CT+ ≈ CT +* > CT+sh (Supplementary Figs. 8a, 2d, e). Thus, the markedly improved stability of the CT+ variants on membranes directly correlated with their improved membrane fission activities. Remarkably, fission efficiency was directly proportional to the preponderance of the assembly-primed, compact dimer conformer in solution in the presence of GTP, sampled almost exclusively by the CT+ variants, but not WT (Fig. 2a). Importantly, contrary to prevailing models, membrane fission activity was inversely correlated with the assembly-stimulated GTP hydrolysis rate on membranes, with CT+ variants of lower GTPase activity being more efficient in fission (Supplementary Fig. 8a, b).

To further assess the impact of the CT-SLiM modifications and imposed structural alterations on membrane remodeling, we used cryo-EM to analyze the self-assembly of WT Drp1 and CT variants on preformed membrane NTs in the constant presence of GTP (Fig. 4c). In agreement with the real-time fluorescence measurements, the cryo-EM data revealed that WT Drp1 formed organized helical polymers that constricted the NTs to a radius of ~15 nm. By contrast, CT+ Drp1 formed disorganized, fuzzy coats that further constricted the membranes to critical radii of <7 nm, frequently resulting in fission and consequent retraction of the cut NTs to the membrane reservoirs located on the EM grid. Interestingly, ΔCT4 Drp1 displayed helical polymers with highly variable diameter (Fig. 4c) consistent with a near complete loss of CT-SLiM-imposed inter-subunit helical register and polymer geometry. Notably, under these conditions, WT Drp1 polymers were observed on both highly curved and relatively flat membrane regions, whereas the CT+ variant was curvature-selective with an acute preference for binding the curved NTs (Fig. 4c).

Together, these data indicate that the CT-SLiM governs both Drp1 polymer geometry and dynamics on membranes, and that CT modifications differentially affect membrane curvature selectivity and fission activity.

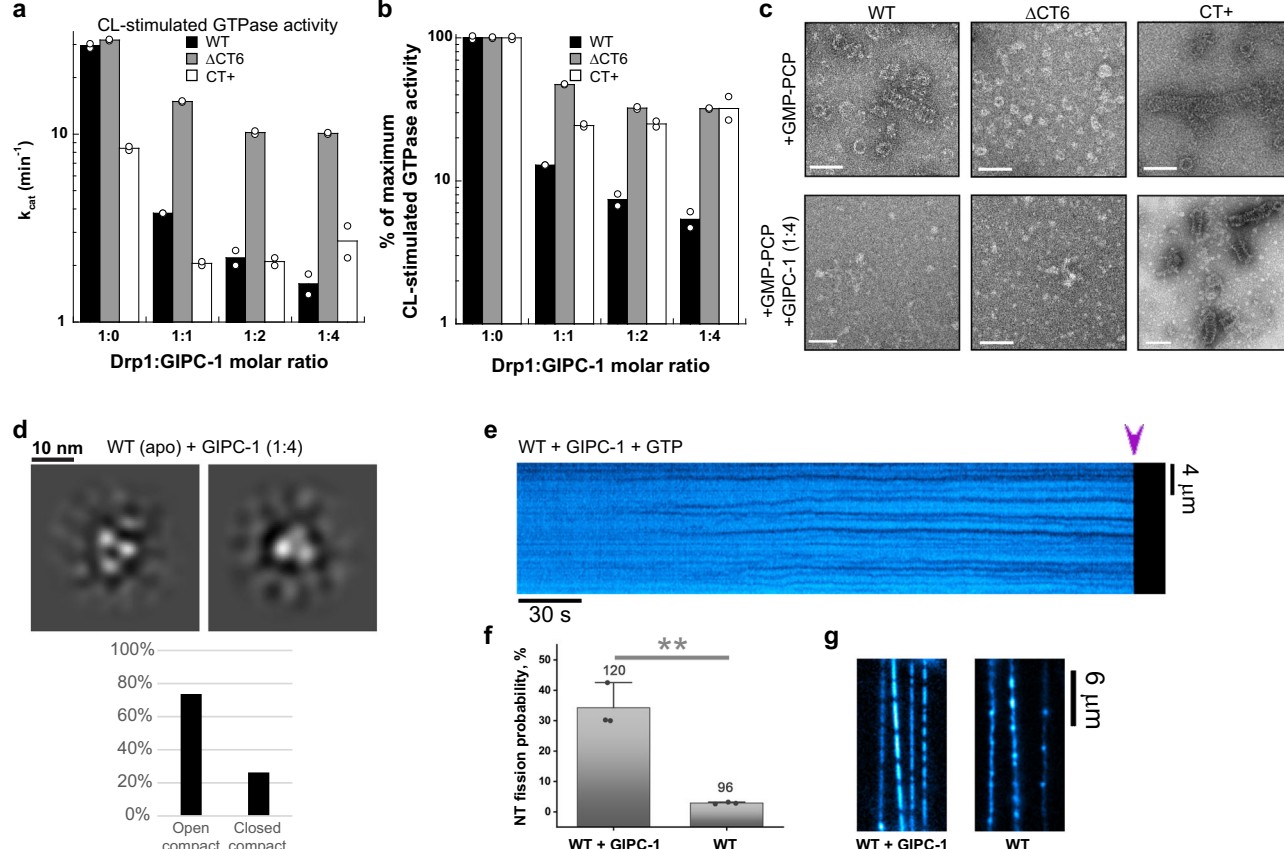

**Fig. 5 | CT-SLiM interactions with GIPC-1 potentiate membrane fission in vitro.**
**a** CL-stimulated GTPase activities of WT and ΔCT6 Drp1 with increasing concentrations of GIPC-1. $k_{cat}$ from two independent measurements are plotted versus Drp1:GIPC-1 molar ratio. **b** Data in **a** replotted as % of maximum activity for each variant. **c** Representative NS-EM images of WT Drp1 and CT variants incubated with GMP-PCP in the absence and presence of a 1:4 molar ratio of GIPC-1. Scale bar, 100 nm. **d** NS-EM 2D class averages of WT Drp1 in the apo state in the presence of a 1:4 molar ratio of GIPC-1. Only the compact conformers (open and closed) were observed for WT-Drp1 in the presence of GIPC-1. **e** Kymograph showing NT constriction and fission (arrowhead) by WT Drp1 in the presence of GIPC-1 at a 1:1 molar ratio (0.5 μM each) in the presence of 1 mM GTP. NT membrane fluorescence is displayed in cyan pseudocolor for clarity. **f** Percentage of NTs that underwent fission upon addition of either 0.5 μM WT Drp1 alone or a 1:1 mixture of WT Drp1:GIPC-1 at 0.5 μM each in the presence of 1 mM GTP. Each point represents a replicate. The number on top of each column represents the total number of NTs for each condition. Mean ± SD are shown. ** Statistically different at the 0.01 level (unpaired two sample *t*-test, equal variance assumed). **g** Images showing the rigid polymerization of WT Drp1 alone versus the formation of much shorter scaffolds in the equimolar presence of WT Drp1 and GIPC-1 on NTs. Images shown were acquired approximately 2 minutes after the addition of the proteins at 2 μM final concentration each in the presence of 1 mM GTP. RhPE channel is shown. Pseudocolor is used for clarity.

## CT-SLiM interactions with GIPC-1 regulate Drp1-mediated membrane fission

Next, we determined how Drp1-GIPC-1 interactions via the CT-SLiM affect Drp1 structure, assembly, dynamics, and function.

GIPC-1 contains an N-terminal IDR in addition to a centrally located PDZ domain flanked by two unique GIPC homology domains (GH1 and GH2)[36] (Supplementary Fig. 9a). The GH1 and PDZ domains are involved in GIPC-1 multimerization[37], whereas GH2 binds MYO6. In the absence of the N-terminal IDR and a PBM (PDZ domain ligand), GIPC-1 forms an auto-inhibited, PDZ domain-swapped dimer that occludes both the PBM and MYO6 binding sites[36] (Supplementary Fig. 9b). However, consistent with a previous report[37], we found that in the unliganded state, and at relatively low concentrations in solution, purified full-length GIPC-1 exists in a fast, dynamic monomer-dimer equilibrium that largely favors monomers (Supplementary Fig. 9c, d). Conversely, at very high concentrations experienced during purification, GIPC-1 also formed long filamentous sedimentable polymers in solution (Supplementary Fig. 10) indicative of its capacity to multimerize when sequestered locally. Purified GIPC-1 at low bulk concentrations in solution however remains soluble (Supplementary Fig. 10) and does not associate with, or remodel, membranes (Supplementary Fig. 10). Together, these data indicated that Drp1 CT-SLiM

binding may function to relieve GIPC-1 auto-inhibition, elicit GIPC-1 multimerization, and promote cooperative Drp1-GIPC-1 co-assembly on membranes. Consistent with this notion, multimeric GIPC-1 has previously been localized to membranes[37] indicating a role for the GIPC-1 multimerization in ligand protein (e.g. Drp1) confinement at target membrane sites.

GIPC-1 robustly inhibited the assembly-stimulated GTPase activity of WT Drp1 on CL-containing membranes in a concentration-dependent manner (Fig. 5a, b). ΔCT6 and CT+ Drp1, by contrast, were modestly inhibited, indicating weakened binding. The modest, but considerable, inhibition for these variants also indicated the presence of additional GIPC-1 interaction sites besides the CT-SLiM (Fig. 5a, b).

GIPC-1 also potently inhibited the GMP-PCP-induced self-assembly of WT Drp1 into rings and spirals in solution in NS-EM experiments, indicating that GIPC-1 binding hinders the helical propagation of Drp1 (Fig. 5c). As expected, GIPC-1 did not considerably affect the GMP-PCP-induced helical self-assembly of the CT+ variant (Fig. 5c). Nevertheless, GIPC-1 still reduced the GMP-PCP-induced formation of triangular nubs by the ΔCT6 variant probably owing to the presence of additional binding sites (Fig. 5c).

Likewise, GIPC-1 potently inhibited the WT Drp1-mediated tubulation of CL-containing liposomes (Supplementary Fig. 10b). ΔCT6

Drp1 membrane remodeling activity, conversely, was not significantly affected (Supplementary Fig. 10c). Interestingly, at equimolar concentrations (1:1) under these conditions, WT Drp1 and GIPC-1 formed amorphous assemblies in solution, whereas at higher GIPC-1 ratios (1:4), linear and bundled filaments of assembled protein in solution, reminiscent of Drp1 co-assembly with the adaptor mitochondrial dynamics protein of 49 kDa or MiD49[38] were evident (Supplementary Fig. 10b). Together, these data indicated that GIPC-1 interactions via the CT-SLiM alters Drp1 self-assembly geometry, with pronounced effects on membrane remodeling as determined by the lack of ordered helical self-assembly and resultant membrane tubulation. A similar inhibition of CL-stimulated GTPase activity and helical self-assembly was observed for WT Drp1 in the presence of GIPC-1 on GalCer CL-NTs indicating that GIPC-1 regulation of Drp1 activity does not vary with membrane curvature (Supplementary Fig. 11a, b).

Surprisingly, NS-EM 2D classification of apo WT Drp1 dimers in the presence of GIPC-1 revealed the presence of the assembly-primed, compact Drp1 conformer in solution (Fig. 5c), in contrast to the auto-inhibited, extended Drp1 conformer predominantly found in GIPC-1's absence (Fig. 2a). Additional density representing GIPC-1, however, was not readily evident reflecting either a dynamic interaction of GIPC-1 with WT Drp1 in the apo state or a substantial overlap of GIPC-1 density with the closely spaced G domains of the compact WT Drp1 conformer. In the case of ΔCT6 and CT+ Drp1, however, various extra densities and altered subunit arrangements were observed attesting to the presence of additional GIPC-1 binding sites and alternate Drp1-GIPC-1 interactions (Supplementary Fig. 11c, d).

The reduced assembly-stimulated GTPase activity observed for Drp1 in the presence of GIPC-1 seemingly potentiates the membrane remodeling events leading to fission, as WT Drp1 in the presence of GIPC-1 catalyzed fission in >30% of free-standing NTs over a wide range of membrane curvatures (Fig. 5e, f, Supplementary Movie 5). Interestingly, at 0.5 μM Drp1 concentration and a 1:1 GIPC-1:Drp1 ratio, we detected the formation of short Drp1 scaffolds on the NTs immediately prior to fission. Besides, these scaffolds were highly mobile on the NT surface (as observed in the kymograph in Fig. 5e, Supplementary Movie 5), suggesting that in the presence of GIPC-1, Drp1 initially assembles into small pre-curved units, and not into complete rings, on the NT surface. Thus, GIPC-1 association with WT Drp1 via the CT-SLiM appears to disengage Drp1 inter-subunit interactions that promote higher-order Drp1 self-assembly.

This differential behavior of WT Drp1 in the presence of GIPC-1 was better evidenced at higher protein concentrations (2 μM, Fig. 5g). Whereas WT Drp1 alone rapidly polymerized into long and rigid scaffolds, rendering kinks in the NTs that precluded membrane fission, in GIPC's presence the growth of the WT Drp1 scaffolds on the NTs was comparable to that of CT+ Drp1 (compare Figs. 5g and 4a), and resulted in NT fission at a similar fission rate to that detected at 0.5 μM protein concentration (Fig. 5f). Thus, WT-Drp1 in the presence of GIPC-1 partially mimics CT+ Drp1, which exhibits reduced GTPase activity and altered CT-SLiM interactions.

### CT-SLiM regulation of Drp1 is critical for mitochondrial and peroxisomal fission in vivo

Drp1 catalyzes both mitochondrial and peroxisomal fission[12,39–41]. Yet, GIPC-1 colocalizes with the mitochondria but not considerably with peroxisomes[18]. To determine whether the CT-SLiM is therefore differentially required for Drp1-catalyzed mitochondrial and peroxisomal fission in vivo, we examined and compared mitochondrial and peroxisomal morphology in Drp1 KO mouse embryonic fibroblasts (MEFs) overexpressing N-terminally Myc-tagged WT and ΔCT4/6 variants, and the C-terminally Myc/FLAG-tagged CT+ variant (Fig. 6a, Supplementary Figs. 12–14). Empty vector-transfected Drp1 KO MEFs displayed extensively hyperfused mitochondria or highly elongated peroxisomes in the absence of Drp1-catalyzed mitochondrial and

peroxisomal fission (Fig. 6a, Supplementary Figs. 12–14). As expected, exogenous Myc-WT Drp1 overexpression effectively rescued and restored both mitochondrial and peroxisomal fission leading to the formation of highly fragmented mitochondria and distinctly punctiform (spherical) peroxisomes (Fig. 6a–d, Supplementary Figs. 12–14). By contrast, however, the overexpression of the ΔCT4 and ΔCT6 variants had no palpable effect on the initial morphology of either organelle (Fig. 6a–d), with the great majority of cells displaying a pronounced perinuclear clustering of hyperfused mitochondria and retaining highly elongated peroxisomes (Fig. 6a–d, Supplementary Figs. 12–14). More surprisingly, the CT+ variant containing the native CT-SLiM sequence was also significantly impaired in the fission of both organelles, albeit to a lesser degree than the ΔCT4/6 variants (Fig. 6a–d, Supplementary Figs. 12–S14). Thus, in spite of their apparent gain-of-function in effecting the fission of model membranes in vitro, the CT variants appeared nevertheless perturbed in effecting organellar fission in vivo. These data further reiterated that the altered self-assembly properties and dynamics of the CT variants as manifested in vitro, and consequent impairments in effector (e.g. GIPC-1) interactions downstream are likely responsible for their organellar fission defects in vivo. Consistent with this interpretation, when overexpressed in Drp1 KO MEFs, the CT+ variant, which forms supramolecular assemblies in vitro (Fig. 1b), constituted large granular puncta in the cytosol indicative of aggregation. In contrast, the WT and ΔCT variants exhibited a more diffuse and homogeneous distribution (Supplementary Fig. 15a, b). In co-immunoprecipitation experiments, neither overexpressed WT Drp1 nor the CT variants co-precipitated with endogenous GIPC-1 (Supplementary Fig. 15c, d) indicating a highly dynamic interaction that could not differentiate the CT variants from WT Drp1 in GIPC-1 binding. These data further suggested that perturbations in GIPC-1 interactions expected of the CT variants are likely secondary to their primary defects/alterations in self-assembly and conformational dynamics.

From these collective data, we conclude that the native CT-SLiM is a critical structural and functional determinant of Drp1-catalyzed mitochondrial and peroxisomal fission, and that its perturbations influence Drp1 function both in vitro and in vivo.

## Discussion

Structural and functional plasticity are two interlinked characteristics of IDRs[10,42,43]. This is best exemplified by the longest and best-recognized IDR in Drp1, the VD, which is involved in multiple protein-protein and protein-lipid interactions via various identified MoRFs and SLiMs[14]. Remarkably, the VD is auto-inhibitory to premature Drp1 self-assembly in solution[14,44,45], while conversely promoting Drp1 self-assembly and function upon partner interactions, specifically with target lipids on mitochondrial membranes[14,17,26,46], thus reflecting the VD's duality and functional diversity. However, the VD and various other IDRs in Drp1 (e.g. the 80-loop) are relatively poorly conserved (Supplementary Fig. 1a) and are subject to extensive tissue- and organism-specific alternative splicing[47], indicating that some of their ascribed functions may not be entirely universal. Here, we demonstrate that the highly conserved CT-SLiM, previously implicated in Drp1 transport[18], is yet another critical, multifunctional 'toggle' that not only governs Drp1 conformational stability and dynamics, but also directs Drp1 self-assembly, assembly geometry, and cooperative GTP hydrolysis to facilitate partner protein-guided membrane constriction and fission.

Our findings have major implications for the understanding of Drp1 function and regulation. We show that the CT-SLiM is critical for ordered Drp1 self-assembly, as oligomerization of the CT variants propagates out-of-register and eventually becomes disordered. The CT-SLiM also directly impacts Drp1 dimer conformational dynamics in solution, with the CT variants more readily sampling the 'assembly-primed' compact conformation in contrast to the CT-SLiM-imposed,

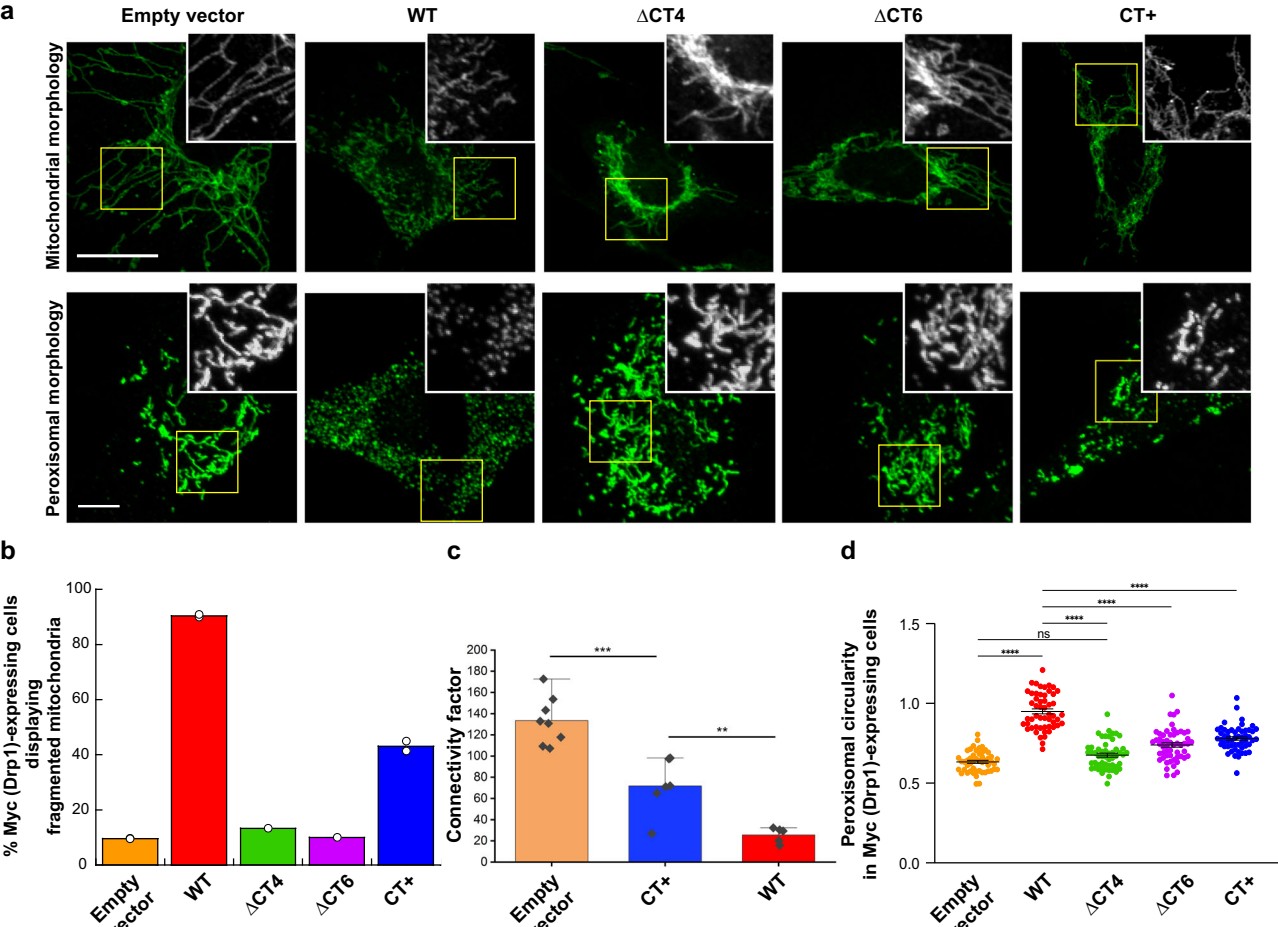

**Fig. 6 | CT-SLiM interactions are essential for mitochondrial and peroxisomal fission in vivo. a** Representative images of mitochondrial and peroxisomal morphologies in Drp1 KO MEFs expressing Myc-tagged WT Drp1 or CT variants. Boxed areas (yellow) are zoomed and shown in grayscale in insets. Individual and merged image color panels are shown in Supplementary Figs. 12 and 14 for mitochondria and peroxisomes, respectively. Scale bar, 20 μm. **b** Percentage of Myc (Drp1)-expressing cells displaying fragmented mitochondria are plotted for WT Drp1 versus CT variants. Total number of Myc (Drp1)-expressing cells analyzed for each variant is $n = 70$ (empty vector), $n = 91$ (WT), $n = 17$ (ΔCT4), $n = 31$ (ΔCT6), and $n = 92$ (CT + ). Individual percentages from two replicate experiments of empty vector, WT, and CT+ are shown. **c** Analysis of mitochondrial network connectivity

as shown in Supplementary Fig. 13d. Connectivity factor is defined as total network length (pixels) per number of separate skeletons detected using the ImageJ skeletonize plugin. Representative cells of $n = 8$ (empty vector), 6 (CT + ), and 5 (WT) from two replicate experiments were examined. Mean ± SD are shown. *** Statistically different at the 0.001 level; ** Statistically different at the 0.01 level (unpaired two sample *t*-test, equal variance not assumed). **d** Plot of peroxisomal circularity in Drp1 KO MEFs expressing WT Drp1 and CT variants. Error bars indicate mean ± SEM. Total number of Myc (Drp1)-expressing cells analyzed for each variant is $n = 50$ (empty vector), $n = 53$ (WT), $n = 53$ (ΔCT4), $n = 55$ (ΔCT6), and $n = 54$ (CT + ). **** Statistically different at the 0.0001 level (one-way ANOVA with Sidak's multiple comparison test). In all such cases, the *p*-values were <10$^{-15}$.

'auto-inhibited' extended conformation of the WT. Consequently, the CT-SLiM emerges as a critical allosteric controller of Drp1 cooperative GTP hydrolysis. On membranes, the CT-SLiM promotes the formation of long, ordered, and curvature-adaptable scaffolds. Yet, at the same time, it restricts scaffold disassembly required for high membrane constriction and fission[48]. Such inherent rigidity is reduced upon CT-SLiM interactions with GIPC-1, which lead to controlled Drp1 scaffold disassembly. Such dynamic rearrangements facilitated by effectors, such as GIPC-1 here, results in the geometric flexibility necessary for membrane constriction and fission, as demonstrated for Dyn1 previously[48].

Interestingly, our results point to an inverse correlation between the GTPase and in vitro fission activities of our Drp1 variants. The CT+ variant, in contrast to WT, forms supramolecular scaffolds in the presence of GMP-PCP. Yet, in the presence of GTP, these variants follow a different pathway. In the case of WT, the energy of GTP hydrolysis powers a robust, but limited, membrane constriction. Whereas, for the CT+ variant, GTP is utilized for scaffold disassembly (melting), resulting in the formation of fuzzy scaffolds, highly sensitive to membrane

curvature. Such apparent scaffold disorganization and flexibility seem to be crucial for membrane fission. Importantly, scaffold flexibility is also present for the ΔCT variants in the presence of GTP. Yet, it is not as pronounced as for the CT+ variant. Consistently, the fission efficiencies of the ΔCT variants are much lower than that of CT+ , while fission is restricted to NTs of the highest initial curvature. The apparent disconnect between in vitro NT fission and GTPase rates for WT and the CT variants can thus be explained by the differential use of the energy of GTP hydrolysis, with the CT-SLiM directing membrane constriction in WT, and disassembly in the already disordered CT variants.

Regardless, neither of the fission-promoting CT variants in vitro were effective in mediating organellar fission in vivo, underscoring the importance of the stringent, allosteric control of Drp1 by the CT-SLiM at target membrane division sites in situ. Although seemingly unexpected, examples of such discrepancies between in vitro and in vivo activities are found in the literature for both Drp1 and classical Dyn1[31,48–50]. Moreover, a lack of correlation between GTPase and self-assembly activities in the regulation of Dyn1 by SH3 domain-containing binding partners has been previously raised[51]. In Dnm1p, the yeast

ortholog of mammalian Drp1, and in Dyn1, a conserved K705A mutation in the GED (K694A in Dyn1) that impairs GTPase activity nevertheless accelerates mitochondrial fission[52] and endocytosis[53]. While a total ablation of GTPase activity impairs function in both Drp1[46] and Dyn[54], a regulated deceleration of GTPase activity mediated by effectors, such as GIPC-1, are seemingly utilized to either promote or inhibit fission.

To reprise, we demonstrate that the native CT-SLiM is an essential intra- and inter-molecular interaction motif that not only governs Drp1 subunit conformational dynamics and oligomerization propensity, but also functions as a spacer that directs Drp1 self-assembly and propagation in the proper helical register. In addition, the CT-SLiM also functions as an auto-inhibitory motif, which in the absence of alleviating binding partners such as GIPC-1, restricts high membrane curvature generation (superconstriction) in order to control and enable partner protein-regulated membrane fission. Based on its composition ($^{694}$RETHLW$^{699}$), we surmise that a combination of electrostatic and hydrophobic interactions mediated by its highly conserved N- (via R$^{694}$ and E$^{695}$) and C-termini (via L$^{698}$ and W$^{699}$), respectively, facilitates this critical role. Interestingly, T$^{696}$ nested within this motif is thought to be a potential site for phosphorylation by Ser/Thr kinases[18,55] already known to modify the disordered VD at various locations[56–58] to regulate Drp1 function.

Remarkably, the Drp1 CT-SLiM, though structurally disparate, functionally parallels the much longer yet disordered C-terminal proline-rich domain (PRD) of prototypical dynamin in its regulatory capacity[59,60], albeit in distinctive ways. Like the CT-SLiM, the PRD binds partner proteins essential for dynamin-catalyzed membrane fission in vivo[61,62]. Similar to that of the Drp1 CT-SLiM (ΔCT4/6), deletion of the PRD in dynamin (ΔPRD) results in increased GTPase activity[14]. These data indicate that the unpartnered CT-SLiM, much like the PRD, functions as a negative regulator of inter-subunit G domain-dimerization necessary for cooperative GTPase activity. Correspondingly, CT+ Drp1, containing a mini-PRD-like non-native extension, duplicates dynamin[48,63] in mediating the fission of model membranes independent of protein partners or receptors in vitro[48,63], a phenomenon not evident with WT Drp1 in our assay setup under our experimental conditions[17]. These data suggest that an effector-induced dampening of the GTP hydrolysis rate and/or increased residence time of the GTP- or transition state-bound Drp1 oligomer on the membrane, instead of robust GTP hydrolysis and rapid oligomer disassembly, potentiates membrane-remodeling leading to complete membrane fission. In this regard, Drp1 conserves the characteristic feature of typical small molecular weight signaling GTPases (e.g. Ras, Rho), which reside alternatively in the GTP-bound, functional "on" and post-GTP hydrolysis GDP-bound, quiescent "off" states, interconverted by GEFs and GAPs, respectively[64,65]. For Drp1, target receptors and lipids likely fulfill these roles. Recent studies show that the extreme C-termini of the distantly related atlastins[66,67], involved in ER membrane fusion, function in a similar auto-regulatory capacity[67,68]. Thus, from an evolutionary standpoint, the extreme C-terminus may represent a critical, conserved, regulatory feature of all dynamin superfamily proteins (DSPs)[66].

Our combined experimental and theoretical modeling data further reveal that the non-native CT+ extension, by virtue of its spurious intra- and inter-molecular interactions, strongly restricts the conformational dynamics of the minimal Drp1 dimer in solution prior to higher-order self-assembly, and artificially increases Drp1 oligomer stability in the presence of GMP-PCP or upon self-assembly on membranes. In addition, given the proximity of the native CT-SLiM to the nucleotide-responsive, dynamically swiveling BSE-stalk interface of an adjacent dimeric subunit in the Drp1 oligomer[15], the non-native CT+ extension likely restricts GTP hydrolysis-dependent BSE-stalk interfacial movements responsible for oligomer disassembly. The non-native CT+ extension may thus conformationally prime Drp1 for

dynamics-resistant oligomerization, consequently enhancing its membrane remodeling capacity, albeit artificially. For WT Drp1, binding partners GIPC-1, which binds the native CT-SLiM[18] and thus restricts BSE-stalk conformational motion, and MiD49/51, which directly binds the BSE-stalk interface[15,38] and alters assembly geometry, likely enable this conformational priming reaction essential for mitochondrial fission. Interestingly, GIPC-1$^{ΔN-IDR}$ is a domain-swapped dimer, whose longitudinal dimension and inter-PDZ domain spacing (~10 nm) closely approximates the distance between the two G domains in the V-shaped compact conformer of the Drp1 dimer[19]. By bridging the two G domains via CT-SLiM interactions, GIPC-1 may function to steer the Drp1 dimer toward the assembly-primed compact conformation. Other contextual partners, such as mitochondrial fission factor (Mff), fission factor 1 (Fis1), and MiD49/51[69], may play similar independent and/or synergistic roles in controlling this Drp1 conformational equilibrium between 'auto-inhibited' and 'assembly-primed' states as previously alluded[16,70].

Finally, we note that the triangular nubs formed by the ΔCT4 and ΔCT6 variants in presence of GMP-PCP in solution are highly reminiscent of the triangular arrangement of WT Drp1 dimers in presence of GMP-PCP and MiD49 observed in the aforementioned cryo-EM study[15]. Similarly, the linear fibrils of Drp1 observed in the presence of excess GIPC-1 are evocative of the linear copolymers of Drp1 and MiD49[38] formed under similar conditions. These observations reaffirm our notion that partner protein interactions steer and direct Drp1 inter-subunit spacing, oligomerization geometry, nucleotide-sensitive conformational rearrangements, and assembly-disassembly dynamics, and are thus indispensable for physiologically relevant Drp1-catalyzed mitochondrial fission.

In summary, our data demonstrate that the native, disordered CT-SLiM is an essential structural and mechanistic determinant of Drp1 function in mitochondrial and peroxisomal division.

## Methods

### Protein production

Human Drp1 (isoform 3) WT, ΔCT4, and ΔCT6 subcloned in pRSET C (Invitrogen), CT+ subcloned in pET21b (Novagen), and CT+* subcloned in pET Biotin His$_6$ FLASH (Addgene Plasmid #30184) were expressed and purified using a combination of His$_6$-affinity and ion exchange chromatography as previously described[26,71]. Mouse CT+ identical to human CT+ in length (699 aa residues) and also in composition except for eight alternative residues within the G domain (3) and VD (5) was obtained from Addgene (Plasmid # 72927)[72] and produced using the same protocol. The CT+ variant was referred to as Drp1-C in our previous study[25]. CT+$^{sh}$ was produced from mouse CT+ by human rhinovirus (HRV) 3 C protease cleavage. For iTFS measurements, single Trp-only mutations (two of the three native Trp mutated to Phe) and single Trp-to-Phe mutations (one of the three native Trp mutated to Phe) were introduced by site-directed mutagenesis in human WT Drp1 subcloned in pRSET C. A non-native Trp present in the N-terminal His$_6$ affinity tag of pRSET C was substituted with Phe in the pertinent constructs used in iTFS experiments. WT Drp1 with a short 7-aa residue N-terminal tag derived from HRV 3 C protease digestion was obtained from the Mears lab and is described elsewhere[73]. For the studies in Drp1 KO MEFs[74], human Drp1 WT, ΔCT4, and ΔCT6 were subcloned in pCMV-Myc (Clontech) and expressed with an N-terminal c-Myc epitope tag, whereas human CT+ was subcloned in pCMV6 (Origene), which conversely appended tandem c-Myc and FLAG epitope tags at the C-terminus. GST-tagged mouse GIPC-1 (sourced from Addgene (Plasmid #35791)) subcloned in pGEX-6P1 (Cytiva) was expressed and purified using standard protocols. The N-terminal GST tag was removed post-purification by HRV3C proteolysis. Insoluble protein aggregates formed during GIPC-1 production as previously noted[36] were removed by high-speed centrifugation at 20,000 × g for 30 min at 4 °C and/or by gel filtration over a Bio-Rad SEC650 column at 4 °C,

and were subsequently identified to be composed of higher-order GIPC-1 oligomers and filaments formed at high protein concentrations. All proteins were stored in buffer A (20 mM HEPES, pH 7.5, 150 mM KCl) containing 1 mM DTT and 10% (v/v) glycerol.

## Liposome and GalCer CL-NTs production

All lipids were obtained from Avanti Polar Lipids Inc. Liposomes containing 25 mol% bovine heart cardiolipin (CL), 35 mol% dioleoylphosphatidylethanolamine (PE), and 40 mol% dioleoylphosphatidylcholine (PC) were prepared in buffer A by extrusion through 400-nm pore-diameter polycarbonate membranes and used in NS-EM and CL-stimulated GTPase assays as described earlier[26]. Rigid lipid NTs composed of 25 mol% CL, 35 mol% PE, and 40 mol% C24:1 β-D-galactosylceramide (GalCer) were prepared using a sonication protocol as described elsewhere[75], and used for negative-stain EM experiments.

## SEC-MALS

SEC-MALS analysis was performed as previously described[26]. Briefly, WT Drp1 and mutants at the indicated injection concentrations were sieved through a Superose 6 10/300GL column attached to an ÄKTA-pure FPLC system (Cytiva) connected in line with DAWN Heleos-II 18-angle MALS and Optilab T-rEX differential refractive index (dRI) detectors from Wyatt Technology. Full-length GIPC-1 (10 μM at injection) was sieved using a Superdex 200 10/300 GL column similarly. Data were analyzed using the ASTRA 7 software also from Wyatt Technology.

## NS-EM and data processing

Negative-stain samples were prepared using 2% (w/v) uranyl acetate (Polysciences, Inc.) on carbon-coated grids as previously described[26].

For the analysis of oligomeric ring structures in the presence of GMP-PCP, 2 μM Drp1 was incubated with 1 mM GMP-PCP in buffer A containing 2 mM MgCl$_2$ and 1 mM DTT final for 30 minutes. Samples were imaged on a Tecnai T12 (FEI Co.) electron microscope at 120 keV, and 10-15 images were acquired using a Gatan 4k × 4k camera at a magnification of 49,000x. For analysis of dimer single particles classes, the apo (Drp1 at 2 μM), +GTP (1 mM), and +GIPC (8 μM) samples were imaged on a TF-20 FEG electron microscope (FEI Co.) operating at 200 kV and recorded at 50,000x magnification with a Tvips Tietz 4k × 4k CMOS-based camera to collect 200 micrographs for each condition.

Data processing was done in cryoSPARC[76]. CTF correction was done using Patch CTF. For all conditions, ~100 particles were manually selected to create an initial picking template for automated picking. For each unique 2D class average identified, individual templates were selected, and all samples were searched using those templates to determine if those class averages were also represented in these samples, even if it was not the predominant class average. The GMP-PCP samples' initial particle stacks were 10,000–20,000 particles, after one round of classification, while final stacks were 1500–6000 particles. The single particle samples' initial particle stacks were 300,000–400,000 and required several rounds of 2D classification (2-4 iterations) to sort through the low SNR associated with small particles. Final stacks were 55,000–180,000 particles.

The number of particles in the extended 'E' and compact 'C' conformations in the *apo* and +GTP states were: WT (apo) – 144k (E) and 36k (C); WT + GTP – 106k (E) and 7.9k (C); ΔCT4 (apo) – 57k (E) and 130k (C); ΔCT4 + GTP – 57k (E) and 67k (C); ΔCT6 (apo) – 28k (E) and 44k (C); ΔCT6 + GTP – 62k (E) and 55k (C); CT+ (apo) - 0 (E), 11k (open C), and 112k (closed C); and CT+ plus GTP – 27k (E) and 76k (C). For +GMP-PCP, the number of particles found in the wide ring (WR) and collapsed ring (CR) morphologies were: WT - 2.2k (WR) and 0.7k (CR); ΔCT4 - 0.7k (WR) and 6.8k (CR); ΔCT6 - 0.6k (WR) and 3.9k (CR); and CT+ - 1.2k (WR) and 0.1k (CR).

Class averages for the WT 'E' (73,700 particles) and ΔCT6 'C' (44,000 particles) conformations were further refined using cryoSPARC's ab initio classification and 3D homogenous refinement. The Drp1ΔVD crystal structure (PDB ID: 4BEJ) was docked using rigid body docking in Chimera. To dock within the compact volume, G Domains were separated, manually repositioned to fit within the volume, and reconnected to the stalks. Idealized models were generated from the docked structures by thresholding the structures to 40 Å in Chimera. These volumes were imported into cryoSPARC. Templates were created to generated idealized 2D class averages from the imported volumes.

For negative-stain imaging on membranes, Drp1 (2 μM final) was incubated with CL-containing liposomes or GalCer CL-NTs (50 μM final total lipid) for 30 minutes in buffer A containing 1 mM DTT. MgCl$_2$ (2 mM final) and either GMP-PCP or GTP (1 mM final) was added to GalCer CL-NT-preincubated Drp1 samples and maintained for an additional 30 min prior to staining. For Drp1 experiments with GIPC-1 on liposomes and GalCer CL-NTs, Drp1 (1.5 or 2 μM final) was pre-incubated with GIPC-1 (at 2, 6 or 8 μM final for 1:1 and 1:4 Drp1:GIPC-1 ratios) for 15 min at room temperature before addition of lipids (50 μM final total) and incubation for a further 15 min.

Membrane tube and oligomeric ring diameters were determined as previously described[77]. Briefly, a broad sampling of rings/helices/tubes were imaged throughout the grid. Measurements were made in ImageJ (NIH) and distributions were generated using either Microsoft Excel or Graphpad Prism.

## SEC-SAXS

SAXS data were acquired at the BioCAT beamline at Sector 18ID of the Advanced Photon Source (APS) at the Argonne National Laboratory, Chicago, USA. Dimeric R403A Drp1 at ~2.8 mg/ml (~34 μM) in 300 μL of buffer A containing 1 mM DTT was centrifuged at 16,000 × g for 10 min at 4 °C to remove any particulate matter. The clarified sample was then sieved through a Superdex 200 10/300GL increase column connected to an autosampler with continuous uni-directional flow at a flow rate of 0.6 mL per min. The scattering intensity data were acquired with 1 sec exposure as previously described elsewhere[78]. SEC-SAXS data were processed using the program BioXTAS RAW[79–81] and PRIMUS in ATSAS package[64]. The scattering curves were first analyzed for aggregation using the Guinier region. The forward scattering I(0), and the radius of gyration, R$_g$, were computed using the Guinier approximation. R$_g$ provides a measure of the overall size of the macromolecule. The pair distance distribution function P(r) was computed from the extended scattering patterns using the indirect transform program GNOM in PRIMUS. The maximum dimension of the particle, $D_{max}$, was estimated from the P(r) function satisfying the condition P(r) = 0. The molecular folding and compactness of the protein were analyzed using the normalized Kratky plot. A bell-shaped profile from the scattering pattern in a normalized Kratky plot is indicative of a compactly folded protein, whereas a plateau at high scattering values is indicative of high flexibility or disorder. Ab initio 3D envelope reconstruction of R403A Drp1 was obtained using the DAMMIN/DAMMIF module in PRIMUS from ATSAS package. The SAXS-derived low-resolution 3D structure was superimposed with the X-ray structure of the ΔVD dimer (PDB: 4BEJ) using Chimera[82,83].

The elution and scattering profiles as well as the acquired scattered intensity plot of R403A Drp1 (Supplementary Fig. 4b, c) indicated that the sample was free of aggregates. The linear fit of the Guinier region (Supplementary Fig. 4d) further indicated sample homogeneity. Interestingly, the radius of gyration (R$_g$) calculated from the Guinier plot corresponded to 60.74 ± 1.52 Å and was indicative of a compact conformation. The value for $D_{max}$, determined from interatomic pair distance distribution function or the P(r) curve (Supplementary Fig. 4e), via indirect Fourier transform of the scattered

intensity, was ~230 Å, conversely pointing to an extended conformation consistent with the ΔVD Drp1 dimer crystal structure. The diversity in Drp1 dimer conformation was further evidenced by examining the normalized Kratky plot (Supplementary Fig. 4f), which showed bell-shaped profiles representative of a compactly folded globular protein as well as plateau at larger scattering values indicative of a high degree of protein flexibility.

## Computational 3D model prediction

The atomic models of the different Drp1 constructs in their monomeric forms were calculated using AlphaFold version 2.1.1[84] running on the Viking Cluster (University of York), using templates from PDB structures with date of deposition up to 14 May 2020. Multiple Sequence Alignments (MSA) were run on the full sequence databases ('--db_preset=full_dbs'). 8 CPUs and two CUDA-enabled Graphics Processing Units (GPU) were used for each job. Five models were produced by default for each construct; the one with highest average pLDDT was taken. The compatibility of the produced models was tested and illustrated by superposing the constructs onto the cryo-EM structure of human Drp1 (PDB ID: 5WP9, EMDB map EMD-8874) using the GESAMT software[85] of the CCP4 suite[86]. The calculated AlphaFold models showed varying degrees of predicted accuracy (average ± SD): $82.46 \pm 10.37$ for C-terminally tagged CT+ Drp1 and $64.24 \pm 21.51$ for N-terminally tagged WT Drp1 in which the tag remained mostly disordered.

## iTFS

iTFS measurements were performed in a $4 \times 4$ mm quartz cuvette (Starna Cells, Inc., Atascadero, CA) at 25 °C using a Fluorolog 3-22 photon-counting spectrofluorometer (Horiba Jobin Yvon) equipped with a 450-W xenon lamp, double-excitation and double-emission monochromators, a cooled PMT housing, and a temperature-controlled cuvette compartment. WT Drp1, CT +, and site-directed mutants were diluted to 0.5 μM final in buffer A containing 1 mM DTT. Buffer background- and instrument-corrected Trp fluorescence spectra were obtained by selectively exciting Trp at 295 nm (2-nm bandpass) and emission monitored at 1-nm increments between 315 and 415 nm (2-nm bandpass). Data are averages of three scans for each sample.

## GTPase assay

Basal and CL-stimulated GTPase activities were determined using a malachite green-based colorimetric assay as described previously[26]. For CL-stimulated activities, Drp1 (0.5 μM final) was preincubated with either CL-containing liposomes or GalCer CL-NTs (150 μM total lipid final) for 30 min at room temperature before GTP addition and the monitoring of inorganic phosphate release at 37 °C. Likewise, for experiments with GIPC-1, Drp1 and GIPC-1 were premixed and incubated for 15 min at room temperature prior to the incubation of the mixture with CL-containing membranes for a further 15 min before GTP addition. Basal GTP hydrolysis rates were measured with Drp1 alone in the absence of lipids.

## Co-sedimentation assay

WT Drp1 and CT variants at 1 μM protein final were incubated with or without GalCer CL-NTs (100 μM lipid final) in a volume of 50 μL for 30 min at room temperature in buffer A containing 1 mM DTT. MgCl2 (2 mM final) and GTP (1 mM final) were subsequently added to lipid pre-incubated samples and maintained for an additional 30 min at room temperature. Supernatant (S) and pellet (P) fractions were obtained by high-speed centrifugation of the samples at $20,800 \times g$ in a refrigerated microcentrifuge maintained at 4 °C. Densitometry of the S and P fractions after SDS–PAGE of equivalent volumes of each sample and staining with InstantBlue® Coomassie Protein Stain (abcam) was performed using ImageJ (NIH).

Likewise, GIPC-1 at 5 μM protein final was incubated with GalCer CL-NTs (500 μM lipid final) prior to centrifugation and sedimentation analysis.

## NT fission assay

Fluorescently labeled lipid NTs were made as previously described[35]. Briefly, 40 μm silica beads covered by hydrated membrane lamella of the desired membrane composition (PC:PE:CL:RhPE 55:29.5:15:0.5 mol %) were mechanically rolled on top of a SU8 micropillar array manufactured on the cover slip surface of a microfluidic chamber. The chamber was initially perfused with buffer A containing 2 mM MgCl2, 1 mM EDTA, 1 mM DTT, and 0.5 mM n-propyl gallate final. Upon rolling of the beads, small membrane reservoirs formed on top of the pillars were interconnected by freely suspended NTs. The protein(s) of interest was/were perfused into the microfluidic chamber in the presence or absence of 1 mM GTP final in the same buffer. Drp1 and GIPC-1 were premixed at an equimolar ratio immediately before perfusion into the chamber.

NT remodeling and fission were monitored using an inverted fluorescence microscope (Nikon Eclipse Ti, Japan) equipped with a 100X/1.49NA objective lens, a CoolLed pE-4000 light source at a low (10%) intensity, and a Zyla 4.2 sCMOS camera (Andor, Ireland). μManager software was used for image acquisition. Image processing (background subtraction and kymograph building) and statistical analysis were performed using Fiji package in ImageJ[87] and OriginPro 8.0 software, respectively.

NT radii were measured following the previously developed protocols based on membrane area to fluorescence correlation[35,88]. Briefly, supported lipid membranes with the same lipid composition as the NTs were formed on a plasma-cleaned cover glass either by spreading of a membrane supported on a silica bead or through bursting of giant vesicles. The resulting flat membranes were thoroughly washed with buffer A. Images of supported membranes were then acquired with the same light intensity and camera setting as the ones used for the NT fission assays. The ROIs' integrated fluorescence intensity was then plotted against the ROIs' area to find the density of the membrane fluorescence signal ($\rho$). The radii of the NTs were obtained from the total fluorescence per unit length ($Fl$) as $r_{NT} = \frac{Fl}{2\pi\rho}$.

## Cryo-EM of Drp1 on preformed NTs

The NTs were produced directly on the glow-discharged Quantifoil R 2/2 300 mesh copper grid. A 2 μL drop of buffer B (10 mM HEPES, pH 7.5, 150 mM KCl) containing 1 mM MgCl2 and 1 mM GTP was placed on the grid. NTs were formed by rolling lamella-covered silica beads as described above for NT formation on micropillars[35]. The resulting NTs were attached to the edges of the holes in the Quantifoil film. Upon NT stabilization, 2 μL of the protein of interest was added to the NTs-containing grid (final protein concentration 0.5 μM). Upon 5 min incubation of the NTs with protein, the excess liquid was removed by blotting with an absorbent filter paper on both sides of the grid for 2 seconds, using a Vitrobot system (Thermofisher) set at 18 °C and 90% humidity. Subsequently, the sample was abruptly vitrified by plunging into liquid ethane (−184 °C). The vitrified grids were maintained in liquid nitrogen and visualized on a JEOL JEM-2200FS/CR microscope equipped with a field emission gun operated at 200 kV and an in-column Ω energy filter. During imaging, non-tilted, zero-loss 2D images were recorded under low-dose conditions, utilizing the 'Minimum Dose System (MDS)' of Jeol software, with a total dose on the order of 30-40 electrons/Å$^2$ per exposure and at defocus values ranging from 1.5 to 4.0 μm. The microscope's in-column Omega energy filter helped us record images with an improved signal-to-noise ratio (SNR) by zero-loss filtering, using an energy-selecting slit width of 20 eV centered at the zero-loss peak of the energy spectra. Digital images were recorded in linear mode on a 3840 × 3712 (5 μm pixels) Gatan K2 Summit direct detection camera (Gatan Inc.) using DigitalMicrograph™ (Gatan Inc.) software, at nominal magnifications of 2000× and 25000×, with a pixel

size of 1.6 nm and 0.154 nm respectively. Images were subsequently treated and analyzed using Fiji software[87].

## Cell biology

Drp1 KO MEFs were plated on 0.1% gelatin (G1393, Sigma)-coated coverslips in 12-well plates. 24 hr after plating, the cells were transfected with the vectors expressing the indicated Myc-tagged Drp1 constructs by using the *TransIT*®-2020 transfection reagent (MIR 5404, Mirus). 24 hr after transfection, the cells were fixed with 4% PFA in PBS for 20 min at room temperature before 5 min permeabilization with 0.1% Triton-X-100. Cells were then incubated with blocking buffer (5% normal goat serum (31872, Thermofisher Scientific), 0.05% Triton-X-100 in PBS) for 1 hr at room temperature. Primary antibodies against Myc (Drp1), Tom20 (mitochondria), and PEX14 (peroxisomes) were incubated overnight at 4 °C. The secondary Alexa-488-labeled goat anti-rabbit (A11034, ThermoFisher Scientific, 1:1000 dilution) and Alexa-568-labeled goat anti-mouse (A11031, ThermoFisher Scientific, 1:1000 dilution) antibodies were incubated for 1 hr at room temperature. The expression of Myc-tagged Drp1 and mitochondrial and peroxisomal morphologies were examined by using anti-Myc (sc-40, Santa Cruz, 1:1000 dilution), anti-Tom20 (11802-1-AP, Proteintech, 1:1000 dilution), and anti-PEX14 (10594-1-AP, Proteintech, 1:1000 dilution) antibodies, respectively. Confocal images were obtained using a 60X oil-immersion objective mounted on an Olympus Fluoview 1000 or 3000 confocal microscope and analyzed by Fiji-ImageJ (NIH).

Western blot analysis of Drp1 overexpression was performed as previously described[26]. Briefly, equal amounts of protein (25 µg) from the total lysates of each transfected cell culture sample, quantified using the Bio-rad protein assay, were resuspended in Laemmli buffer, separated using SDS-PAGE, and transferred onto nitrocellulose membranes. Membranes were probed with anti-Myc (sc-40, Santa Cruz, 1:1000 dilution), anti-Flag (F1804, MilliporeSigma, 1:5000 dilution), anti-actin (A1978, MilliporeSigma, 1:10000 dilution), anti-Drp1 (611113, BD Bioscience, 1:2000 dilution) or anti-GIPC-1 (sc-271822, Santa Cruz, 1:500 dilution) as indicated using the manufacturer's recommendations and visualized by enhanced chemiluminescence.

For co-IP experiments probing Drp1-GIPC-1 interactions, cells were lysed in total cell lysate buffer (50 mM Tris-HCl, pH 7.5, 150 mM NaCl, 1% Triton X-100, and protease inhibitor cocktail). Total lysates were incubated with the indicated antibodies (1:1000 dilution) overnight at 4 °C, followed by the addition of protein A/G beads (sc-2003, Santa Cruz, 20 µL per 1 mL of lysate) for 2 hr at room temperature. The immunoprecipitants were washed four times with cell lysate buffer and analyzed by western blotting using the indicated antibodies.

## Quantification of mitochondrial connectivity and peroxisomal circularity

For examining mitochondrial connectivity, images of cells expressing Myc (Drp1) were selected based on the plot profile maximum of the Alexa Fluor 568 (red) channel, set to 120 ± 20 a.u. For cells expressing the empty vector, the selection was based on Alexa Fluor 488 (green) intensity, with the maximum in 100 ± 0 a.u. The selected images were analyzed with an in-house Fiji[87] macro consisting in four main steps: i) background subtraction, ii) binary mask creation, iii) skeletonize plugin[89] and iv) analyze skeletons command. The resulting dataset was further analyzed using OriginPro software. Branch length was used to calculate the total network length and the average branch length. For the empty vector, all branches below 4 pixels were excluded from the analysis. For the number of skeletons, all skeletons below 4 pixels were deleted for the empty vector. All skeletons with length of 0 pixels were excluded from analysis. The connectivity factor was obtained by dividing the number of skeletons by the total network length.

For examining peroxisomal circularity, images of cells expressing Myc (Drp1) were selected based on the plot profile maximum of the Alexa Fluor 568 (red) channel, set to 50 ± 20 a.u. Images were analyzed

with an in-house Fiji[69] macro "Mitochondrial Analyzer" conducting similar steps as above, to quantify mean peroxisomal area and perimeter. The resulting datasets were analyzed further using GraphPad Prism 10 software. Mean area and mean perimeter was used to calculate peroxisomal circularity via the equation: circularity = $4*\pi*$mean area/mean perimeter$^2$. A value of 1.0 indicates a perfect circle, while those approaching 0.0 indicate increasingly elongated shapes.

## Reporting summary

Further information on research design is available in the Nature Portfolio Reporting Summary linked to this article.

## Data availability

Previously published high-resolution Drp1 structures referred to in our manuscript are readily accessible from the protein data bank (PDB) via accession codes: 4BEJ and 5WP9. The particle numbers analyzed for generating 2D class averages and their relative percentages are noted in Methods. Source data are provided with this paper.

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

## Acknowledgements

We thank Ashutosh Prince and Shane Wyborny (both of CWRU) for technical assistance in protein production. NIH R01 grants GM121583 and GM125844 supported work in the R.R. and J.A.M. laboratories, respectively. Work in the A.V.S. laboratory was supported by the PGC2018-099971-B-I00 and PID2021-127844NB-I00 grants funded by MCIN/AEI/10.13039/501100011033/ and by "ERDF A way of making Europe" and by the Basque Government Grant IT1625-22. Jon Agirre is a Royal Society University Research Fellow (award codes UF160039 and URF\R\221006). I.P.J. acknowledges the predoctoral fellowship from the University of the Basque Country. We are grateful for computational support from the University of York High Performance Computing service, Viking and the Research Computing team, notably Jasper Grimm and Emma Barnes. The authors are grateful to the Electron Microscopy and Crystallography platform of the CIC bioGUNE and the Basque Resource for Electron Microscopy (BREM) of the Biofisika Institute for providing access to cryo EM sample preparation and analysis equipment. Molecular graphics images were produced using the UCSF Chimera package from the Resource for Biocomputing, Visualization, and Informatics at the University of California, San Francisco (supported by NIH P41 RR-01081). This research used resources of the Advanced Photon Source, a U.S. Department of Energy (DOE) Office of Science User Facility operated for the DOE Office of Science by Argonne National Laboratory under Contract No. DE-AC02-06CH11357. BioCAT was supported by grant P30 GM138395 from the National Institute of General Medical Sciences of the National Institutes of Health.

## Author contributions

Conceptualization: A.V.S, and R.R.; Experimental methodology: A.V.S, R.R., J.A.M.; Experimental investigation and analysis: all authors; Writing—original draft: A.V.S, and R.R; Writing—review and editing: all authors; Funding acquisition: A.V.S and R.R.; Resources: J.A.M., A.V.S. and R.R.; Project supervision: A.V.S. and R.R.; Negative-stain EM data acquisition and Drp1 structure reconstructions: K.R. and J.A.M.; Cell biology experiments and peroxisomal circularity analysis: D.H. and X.Q.; SEC-SAXS experiments and data analyses: M.M.; Production of DNA constructs: P.M. and R.R.; Protein purification and GTPase assays: P.M., J.O.G., and R.R.; Alphafold predictions and structural overlays: J.A.; SEC-MALS experiments: R.R.; Analysis of mitochondrial fragmentation: R.R. and A.V.S; Preliminary data on Drp1-mediated NT fission: J.M.M.G.; NT fission assays and analyses: I.P.J. and A.V.S.; cryoEM analyses of preformed NT constriction: I.S.P. and A.V.S.

## Competing interests

The authors declare no competing interests
