## [Peer Review File · Nature Communications]

Allosteric control of dynamin-related protein 1-catalyzed mitochondrial and peroxisomal fission through a conserved disordered C-terminal Short Linear MotifReviewers' Comments:

Reviewer #1:

Remarks to the Author:

Drp1 is a mechanochemical GTPase required for mitochondrial and peroxisomal fission. Drp1 assembles into rings and helices that encircle mitochondria/peroxisomes and a GTP-dependent mechanism leads to constriction and fission. Drp1 is cytosolic and must be recruited to target membranes by integral protein adaptors such as Mff, or MiD49/51. Recent reports indicate that Drp1 is moved to the perinuclear space by an interaction with GIPC1, mediated by the highly conserved C-terminus of Drp1, which contains a short linear motif (SLiM). Due to its intrinsic disorder, the CTD of Drp1 is not present or resolved in high resolution structures and its function has not been tested outside of the GIPC1 interaction. This study analyzes functions of Drp1 C-terminal variants and find that the CT-SLiM is critical for allosteric regulation of Drp1. In addition to the discovery that the Drp1 CT-SLiM is required for Drp1 mitochondrial fission activity in cells, the data presented here reveal that Drp1 GTPase activity is moderated by the CT-SLiM domain. Removal of this self-regulatory region through truncation or extension alters Drp1 GTP driven self-assembly and mitochondrial fission activity. Analysis of purified Drp1 assembly and GTPase activity indicates that truncation of the C-terminus has a distinct effect compared to extension of the C-terminus. While removal of C-terminal residues impaired Drp1 assembly in solution and onto lipid substrate, these truncated variants had higher basal GTP hydrolysis rates compared to wild type. In contrast, the extension variant assembled into longer helices with a similar diameter to wild-type Drp1 and was shown to have decreased rates of GTP hydrolysis. All were loss of function in cells and somewhat surprisingly, each C-terminal Drp1 variant mediates nanotubule fission while the full length, wild-type Drp1 does not. This suggests that the nanotubule fission assay has revealed at least two distinct off-pathway Drp1 assemblies that can mediate fission of nanotubules, but not mitochondria. Addition of GIPC was also inhibitory, decreasing GTPase activity and helical assemblies, but stimulating nanotubule fission.

Overall, the data are convincing and support the conclusion that the C-terminal SLiM motif is not only required to interact with GIPC1, but also to regulate Drp1 self-assembly. The data are somewhat confusing, especially given that the nanotubule fission data are not consistent (everything except Drp1-WT mediates nanotubule fission). More careful discussion throughout would help readers follow the data.

Specific comments:

1. The details of Figure 4 are somewhat confusing, and the following points should be clarified by the authors.

- (i) How was the radius of the nanotubules determined?
- (ii) When authors state that the truncated Drp1 variants selectively mediate fission of highly curved nanotubules, does that mean NTs with a smaller radius or tubes that are also curved in the microfluidic chamber?
- (iii) A more detailed explanation of how the fission probably in Fig 4B was calculated is required.
- (iv) The fact that none of these Drp1 variants support mitochondrial division while all mediate some nanotubule fission raises the possibility that the nanotubule assay allows for fission by a different mechanism than occurs in cells. This should be discussed clearly.

2. The data describing the lipid remodeling activity of Drp1 with GIPC-1 was confusing and should be clarified. Specific points that lack clarity include:

- (i) GIPC-1 impaired Drp1-mediated tubulation of CL-containing liposomes, but increases efficacy of nanotubule constriction and fission. Authors should discuss why the effect of GIPC-1 on liposome tubulation is opposite from its activity on nanotubules.
- (ii) Why was 0.5 micromolar Drp1 used for nanotube fission in Figure 4 while 2 micromolar Drp1 was used in Figure 5? Is Drp1-mediated nanotubule fission altered by GIPC-1 when Drp1 is at 0.5 micromolar?
- (iii) If the 2 micromolar Drp1 concentration results in the formation of non-productive structures, why

is that concentration used for the GIPC-1 fission experiments?

(iv) Authors state that GIPC-1 alters Drp1 inter-subunit interactions that promote extensive polymerization, which is consistent with data in Figure 5B, but also observe quite significant areas of constricted nanotubules in the fission assay, which would require extensive polymerization. Is GIPC-1 activity different when nanotubules are the substrate?

4. The fact that Drp1-CT+ is not functional in cells is vaguely attributed to "perturbed partner protein interactions" but this variant was excluded from in vitro analysis with GIPC-1. Is CL-stimulated GTPase activity of Drp1-CT+ decreased or unaffected by GIPC-1? Does the presence of GIPC-1 alter the proportion of Drp1-CT+ in the closed compact conformation (solution, apo)?

5. When the Drp1 C-terminal variants are expressed in cells, does the structure of peroxisomes change?

Reviewer #2:

Remarks to the Author:

In the present article, the authors have determined the role of C-terminal intrinsically disordered region CT-SLiM in Drp1, a protein that catalyzes mitochondrial membrane fission. By deleting the C-terminal region they show that the mutants failed to assemble into higher ordered structures or form helical assembly in solution but instead form triangular shaped nubs with smaller dimensions. Truncated mutants displayed higher GTPases activity both in basal and in the presence of lipid conditions, decreased membrane fission activity (in vitro and in vivo) which has not been observed in other dynamin family like proteins.

On the other hand, extending the C-terminal domain by adding non-native residues to the C-terminus, resulted in higher helical assemblies in solution compared to WT, lower GTPases activity and higher membrane fission activity in vitro, compared to the deletion mutants.

By utilizing such defective mutants (deletion of C-terminal region) and gain-of-function mutant (extension of C-terminal region with non-native residues), along with utilizing various techniques/software like; SEC-MALS, Negative-stain EM, 3D model prediction by AlphaFold, Trp Fluorescence spectroscopy, fission assays and Cryo-EM, authors have revealed a novel mechanism of mitochondrial fission by Drp1 mediated through CT-SLiM.

The article is well written, results and figure sections are self-explanatory and very well stated. However, there are various conclusions which are premature and need solid evidence from more structural studies.

Major concerns:

The majority of the conclusions are derived from negative stain EM. The authors need to support their results with more structural information obtained by Cryo-EM or X-ray crystallography. For example, the nub-like structures obtained in the Δ CT4 and Δ CT6 variants show a close proximity of G-domains. Can this proximity be addressed by higher-resolution structural studies or by FRET based experiments or other biochemical methods?

The authors should provide more justification for their focus on the CT+ variant, which is physiologically irrelevant.

In fig 2A, how confident are the authors that the extended and compact conformations observed are not due to the protein deposition on the grid in different orientations? Showing all 2D classes in a supplementary figure could further support their conclusions.

A 3D structure, even with negative stained samples, would be beneficial and if not possible explain why.

In fig 2A, WT exists in an extended state. CT+ on the other hand has more activity in terms of forming longer supramolecular helical assemblies and the ring like structures compared to WT. Still, CT+ exists in a closed compact state instead of an expected extended state as in WT. Can the authors comment on this more in the result section?

CT+ shows more helical assembly on lipids, high fission activity but low GTPase activity compared to WT. An expanded explanation of this result would be beneficial. In addition, a sup-pellet assay in the presence of lipids would define different lipid binding and disassembly properties of each mutant compared to WT.

Fig 3B, CT4 and CT6 mutants show almost comparable GTPases activity compared to WT but the fission probability and assembly properties appear much less in WT. Can the authors comment on the apparent disconnect between GTPase activity, assembly and fission.

Fig 4C, provide the statistics with a plot of number of tubes observed vs diameter of each tube in WT, CT+ and Δ CT4.

Fig S9B, show a control of GIPC-1 + lipid.

Minor changes:

Fig 1C. It would be nice to have a label helical diameter below the figure (first part of 1C).

Fig 2A. Please increase the font of the statistics and add the values in the text.

Fig 2B. Helix of CT+ can be labelled better so that it becomes easier to differentiate. An addition of overlay between CT+ and WT would be better to observe the inward buckling of G domain and the differences observed.

Point-by-point response to reviewer comments

We thank the reviewers for the positive feedback on our manuscript. Below we provide a point-by-point response to each of the reviewer's concerns, whose comments we reproduce verbatim (*italicized in bold*). Our responses follow.

Reviewer 1:

1. The details of Figure 4 are somewhat confusing, and the following points should be clarified by the authors.

(i) How was the radius of the nanotubules determined?

Thanks for pointing this out. NT radii were determined by utilizing a recently introduced method of fluorescence intensity-to-membrane area correlation using a flat membrane patch for calibration as described elsewhere¹⁻³. We have added this description to the corresponding Methods section.

(ii) When authors state that the truncated Drp1 variants selectively mediate fission of highly curved nanotubules, does that mean NTs with a smaller radius or tubes that are also curved in the microfluidic chamber?

We have changed Fig. 4b to clarify this point. The truncated variants selectively mediate the scission of tubes with smaller radii, i.e. the ones with the highest curvatures in the sample. We believe that the new version of the figure illustrates the disposition of the fission reaction towards higher curvatures better.

(iii) A more detailed explanation of how the fission probability in Fig 4b was calculated is required.

As this point was not clear in the previous version of the plot, we have changed the data representation to show the overall distribution of starting tube radii, and whether these underwent fission (or not) for each of the tested Drp1 variants. We hope that the idea of curvature selectivity and fission efficacy for each variant is better transmitted with this new representation. We have also unified the protein concentrations for all the variants, so that the new data reflect the results obtained upon addition of 0.5 μ M of each protein in the presence of GTP. This concentration is physiologically relevant as it corresponds to the estimated cytosolic concentration of Drp1⁴, as well as the concentration at which the catalytic activity (k_{cat}) of WT Drp1 nears saturation⁵.

(iv) The fact that none of these Drp1 variants support mitochondrial division while all mediate some nanotubule fission raises the possibility that the nanotubule assay allows for fission by a different mechanism than occurs in cells. This should be discussed clearly.

In new cellular data presented in Supplementary Fig. 15a, we show that the premature self-assembly and aggregation of the CT+ variant in the cytoplasm (inherently or through dysregulated partner protein association), and conversely, the lack of ordered self-assembly and/or effector binding (including GIPC-1) for the Δ CT variants as previously determined⁶, impair both mitochondrial and peroxisomal fission.

However, in our minimal *in vitro* nanotube (NT) system with mechanically preset high membrane curvatures, a need for such Drp1 self-assembly regulation (on flat vs. curved membranes and/or through partner binding) is obviated. Therefore, it is not surprising that both the Δ CT and CT+ variants, which impose greater curvature on membranes than WT, stochastically progress toward fission due to specific defects in CT-SLiM-imposed ordered self-assembly as well as in the auto-inhibition of high-curvature generation (Fig. 4c). It is important to note however that the Δ CT variants owing to variable curvature generation (due to impaired self-assembly interactions; Fig. 4c, Supplementary Fig. 6a) are significantly more defective in NT fission than the CT+ variant, which by contrast forms supramolecular molecular assemblies and hydrolyzes GTP at a much slower rate (Fig. 3, Supplementary Fig. 6d, e), thereby increasing its membrane residence time. We now elaborate on this in the Discussion section.

2. The data describing the lipid remodeling activity of Drp1 with GIPC-1 was confusing and should be clarified. Specific points that lack clarity include:

(i) GIPC-1 impaired Drp1-mediated tubulation of CL-containing liposomes, but increases efficacy of nanotubule constriction and fission. Authors should discuss why the effect of GIPC-1 on liposome tubulation is opposite from its activity on nanotubules.

Membrane fission requires high local curvature. Liposome tubulation, on the other hand, requires long-range, higher-order self-assembly. It was previously shown that such long structures are fission-impaired for classical dynamins, but disassembly in the presence of GTP promotes fission²⁰. We believe that this is what GIPC-1 does in our experiments, i.e., prevent the formation of long Drp1 scaffolds by counteracting CT-SLiM-directed Drp1 self-assembly (see new Supplementary Fig. 11a, b) while promoting high local membrane curvature requires for fission (see Fig. 5e, g). Please note that the liposome tubulation and the assembly/disassembly experiments on liposomes and NT in the presence of GIPC-1 were performed with Drp1 in the *apo* state (in the absence of GTP). NT fission experiments were performed in the constant presence of GTP. A pronounced dampening of the Drp1 GTP hydrolysis rate by GIPC-1 is expected to retain Drp1 in the GTP-bound state, which promotes scaffold growth. However, this is counteracted by GIPC-1 binding of the CT-SLiM, which instead directs disassembly. Thus, short, dynamic, fission-favoring scaffolds are produced by GIPC-1 to control Drp1-catalyzed membrane fission.

How exactly GIPC-1 functions mechanistically to regulate the Drp1 GTP hydrolysis and assembly-disassembly cycles requires a much deeper investigation, which is beyond the scope of this manuscript. This manuscript focuses on the critical importance of the CT-SLiM and how binding partners such as GIPC-1 influence Drp1 activity without delving into finer mechanistic detail saved for future experimentation. We have discussed the rationale for our interpretations in the revised manuscript.

(ii) Why was 0.5 micromolar Drp1 used for nanotube fission in Fig. 4 while 2 micromolar Drp1 was used in Figure 5? Is Drp1-mediated nanotubule fission altered by GIPC-1 when Drp1 is at 0.5 micromolar?

We have now matched both figures to reflect the fission efficiency and tube constriction over time at 0.5 μ M for all proteins used. We are thankful to the reviewer for the comment, as we

realized that 0.5 μM concentration is more physiologically relevant (as we already pointed out in the answer to the previous questions). Importantly, fission efficiency for Drp1 in the presence of GIPC-1 remains at the same level as with 2 μM , while the scaffolds formed at lower Drp1 concentration were smaller and more dynamic. Thus, Drp1-mediated membrane fission in the presence of GIPC-1 appears to be a “kinetically controlled” process that apparently does not depend on scaffold size. We initially used a higher concentration of proteins to emphasize the qualitative difference in the Drp1 scaffold size (length) observed, with and without GIPC-1 (as shown in Fig. 5, panel g). We believe the new data in Fig. 5 allow us to illustrate these points more effectively.

(iii) If the 2 micromolar Drp1 concentration results in the formation of non-productive structures, why is that concentration used for the GIPC-1 fission experiments?

While it is tempting to conclude that higher protein concentrations lead to defective-fission, in this case, it is difficult to conclude that 2 μM concentration results in the formation of non-productive structures. WT Drp1 remains ineffective in fission at both 0.5 μM and 2 μM , whereas fission in the presence of GIPC-1 is similar under both conditions. As stated in the previous comment, we used 2 μM concentration in the initial version of the manuscript to underline the qualitative difference in the size (length) of the Drp1 scaffolds with and without GIPC-1, which is better observed at higher protein concentrations (now shown only in Fig. 5g to illustrate this point).

(iv) Authors state that GIPC-1 alters Drp1 inter-subunit interactions that promote extensive polymerization, which is consistent with data in Figure 5B, but also observe quite significant areas of constricted nanotubules in the fission assay, which would require extensive polymerization. Is GIPC-1 activity different when nanotubules are the substrate?

Please see the response to point (i) above. GIPC1-driven alteration of Drp1 inter-subunit interactions are better observed at lower protein concentrations now shown in Fig. 5 (please compare kymograph 4a for WT with 5e for WT+GIPC-1, both at 0.5 μM). Higher mobility of the GIPC-1-Drp1 scaffolds on the tube as well as their limited extension reveal that scaffold elongation/growth is inhibited in the presence of GIPC-1. At higher protein concentrations, we observe that increasing the GIPC-1:Drp1 ratio has a negative effect on both scaffold polymerization and NT fission (data not shown). Please note that the data in Fig.5c (previously Fig. 5b) reflect the 1:4 Drp1:GIPC-1 ratio, while these proteins are maintained in a 1:1 ratio for the NT fission assay.

4. (i) The fact that Drp1-CT+ is not functional in cells is vaguely attributed to “perturbed partner protein interactions” but this variant was excluded from in vitro analysis with GIPC-1. Is CL-stimulated GTPase activity of Drp1-CT+ decreased or unaffected by GIPC-1?

In the revised manuscript, we have included data for the CT+ variant with GIPC-1 in Fig. 5a-c. Like for the ΔCT6 variant, GIPC-1 also suppresses the CL-stimulated GTPase activity of the CT+ variant, but not to the same degree as observed for WT (Fig. 5b), indicating perturbed GIPC-1 association (reduced affinity) of the CT variants. Consistent with this, EM data presented in Fig. 5c show that whereas GIPC-1 prevents WT Drp1 spiral assembly in the presence of GMP-PCP, the corresponding assembly of the CT+ variant is minimally perturbed. As previously shown,

the triangular nubs formed by the Δ CT6 variant are also impacted, but without the drastic reduction in GTPase activity as seen for the WT. Thus, these data show the Δ CT and CT+ variants both remain significantly perturbed, but not totally ablated, in GIPC-1 association and regulation.

(ii) Does the presence of GIPC-1 alter the proportion of Drp1-CT+ in the closed compact conformation (solution, apo)?

The incubation of the CT+ variant with GIPC without nucleotide results in large-scale aggregation not observed with WT (see below), preventing extensive analyses. However, we successfully processed the single-particle protein in the background using cryoSPARC. Some elongated conformations were now observed (shown in Supplementary Fig. 11c), but as for the CT+ variant alone in the apo state, the compact conformation predominated. In this case, we could not distinguish between closed compact and open compact conformations.

5. When the Drp1 C-terminal variants are expressed in cells, does the structure of peroxisomes change?

We thank the reviewer for this interesting query. GIPC-1, which is predominantly cytosolic, colocalizes with the mitochondria and the plasma membrane but not with the peroxisomes⁶. Interestingly, however, overexpression of the Δ CT and CT+ variants in Drp1 KO MEFs impairs both mitochondrial and peroxisomal fission (Fig. 6a-d). In addition, we found that the CT+ variant forms granular puncta in the cytosol (suggesting aggregation or premature self-assembly) in contrast to a more diffuse, homogeneous distribution of the WT and Δ CT variants. Co-IP experiments with overexpressed Drp1 (bait) and GIPC-1 further revealed that the Drp1-GIPC-1 interaction is highly dynamic and cannot distinguish WT from the CT variants in GIPC-1 binding. Collectively, these data demonstrate that altered structural and assembly properties of the Δ CT and CT+ variants are primarily responsible for their impaired function in cells. Perturbed GIPC-1 (effector) interactions are likely secondary to these effects.

Reviewer 2:

Major concerns

1. *The majority of the conclusions are derived from negative stain EM. The authors need to support their results with more structural information obtained by Cryo-EM or X-ray crystallography. For example, the nub-like structures obtained in the Δ CT4 and Δ CT6*

variants show a close proximity of G-domains. Can this proximity be addressed by higher-resolution structural studies or by FRET based experiments or other biochemical methods?

X-ray crystallography of Drp1, as previously described, entails the excision of the disordered variable domain, which in itself alters the geometry of the dimeric assembly subunit as well as of Drp1 self-assembly⁷⁻⁹. FRET-based experiments (either at the ensemble or single-molecule level) necessitates the validation of a Cys-less but functional Drp1 variant to introduce single Cys at strategic locations for fluorescence labeling. As previously shown⁵, mutation of some of the native Cys residues (specifically C300 in the G domain) alters Drp1's structural/assembly properties and activity. To circumvent these issues, we instead used small angle X-ray scattering (SAXS) to map the conformational landscape and flexibility of a self-assembly-restricted Drp1 dimer (using a R403A mutation in spatially distant interface 3; Fig. 1a). The envelope structure (surface representation; Fig. 2c) shows ample sampling space between the two G domains of the overlaid crystal structure, which is consistent with dynamically interconverting forms facilitated by rearrangements/swiveling around Hinge 1 and compatible with both the 'extended' and 'compact' conformations. We therefore conclude that Drp1 dynamically samples both conformations in solution, with a greater residence time for WT in the assembly-auto-inhibited extended conformation, and a greater residence time for the CT-SLiM deletion/extension mutants in the assembly-primed compact conformation absent of CT-SLiM auto-inhibition. These data are now presented in Fig. 2c and described further in the text.

2. The authors should provide more justification for their focus on the CT+ variant, which is physiologically irrelevant.

We concur that the CT+ variant is physiologically irrelevant. Still, this variant was employed here to dissect the requirement of the carboxylate moiety of the extreme C-terminal residue from the native recognition sequence (⁶⁹⁶THLW⁶⁹⁹), both of which are essential for high-affinity PDZ domain (GIPC-1) binding¹⁰. Indeed, this turned out to be the case as the CT+ variant resists GIPC-1-mediated disassembly (Fig. 5c). We have added a sentence to this effect at the beginning of the Results section to explain our rationale. Moreover, Drp1 variants with C-terminal tags have been used previously (thankfully sparingly) in the literature and have produced both conflicting^{11,12} and confounding results¹³⁻¹⁵, when compared to N-terminally His-tagged and untagged versions^{5,16}. Our study elucidates why and also cautions the field of their further use.

3. In fig 2A, how confident are the authors that the extended and compact conformations observed are not due to the protein deposition on the grid in different orientations? Showing all 2D classes in a supplementary figure could further support their conclusions. A 3D structure, even with negative stained samples, would be beneficial and if not possible explain why.

We seriously considered this possibility but could not reason why the WT rarely sampled the "compact" view, whereas the CT variants sampled it predominantly. In principle, protein deposition on the grid should not be different based on the absence or presence of a short polypeptide segment (six-residue CT-SLiM or an appended affinity tag) at the C-terminus. We therefore reasoned that these may be different conformations than views. To better support this assessment, we now show the starting 2D classes for WT and CT variants in Supplementary Fig. 3c.

Due to the inherent flexibility of the Drp1 molecule, as also revealed by SAXS, single particle 3D reconstruction from 2D negative-stain images poses a significant challenge. Millions of particles are needed to overcome the heterogeneity. In addition, preferred

orientations observed even when in ice (for cryo-EM) are further exacerbated in negative-stain samples. The resulting 3D volumes are often truncated or are missing domains.

We generated 3D volumes from negative-stain data for both the extended and compact conformations, and docked the available Drp1 dimer crystal structure (top row in Fig. 2b) into these. To achieve a general approximation of the compact conformation, the crystal structure was placed in the density using rigid body docking in Chimera. The G domains were separated, repositioned, and reconnected to the stalks. Because the resolution is so low, an idealized model for both conformations were generated from the crystal structure in Fig. 2b (middle row). This allowed us to back project 2D class averages from the volume in cryoSPARC. The results show that the compact conformation *is never observed* in the population of 2D class averages of the extended conformation, whereas the compact conformation, in a few select orientations, appears to be extended (class 8, 12, 18 of the bottom row right-side in Fig. 2b). This information, in addition to the fact many such projected classes were seen in the negative-stain data gives us confidence that the compact conformation observed with the CT mutants is distinct, and not a different orientation of the extended conformation. SAXS and iTFS data, together with the AlphaFold2 modeling, provide orthogonal support to our conclusions.

- 4. In Fig. 2a, WT exists in an extended state. CT+ on the other hand has more activity in terms of forming longer supramolecular helical assemblies and the ring like structures compared to WT. Still, CT+ exists in a closed compact state instead of an expected extended state as in WT. Can the authors comment on this more in the result section?***

We should have previously stated this in the text, but we have done so now. The 'extended' conformation (sampled predominantly by the WT) is essentially auto-inhibited. It restricts higher-order spiral assembly, whereas the 'compact' conformation (sampled mostly by the CT+ variant) is 'assembly-primed' and supports supramolecular helical self-assembly. The compact conformation, as per our description, is the one that drives higher-order self-assembly and not the extended form.

- 5. CT+ shows more helical assembly on lipids, high fission activity but low GTPase activity compared to WT. An expanded explanation of this result would be beneficial. In addition, a sup-pellet assay in the presence of lipids would define different lipid binding and disassembly properties of each mutant compared to WT.***

A faster rate of GTP hydrolysis likely reflects the faster assembly-disassembly dynamics and GTP loading of the reversibly dimerizing G domain at the helical inter-rung interface during cooperative GTP hydrolysis. In other words, fast disassembly and rapid release of GDP and Pi are critical for prompt GTP loading to support subsequent rounds of GTP hydrolysis. The CT+ variant, while supporting higher-order helical self-assembly through its primed 'compact' conformation and via non-native, non-specific interactions mediated by the CT extension, likely lags in G domain disassembly, GTP loading, and subsequent reassembly.

Consistently, in the Sup (S)-Pellet (P) co-sedimentation experiment shown in Supplementary Fig. 6f, all variants appear to bind lipid membranes (CL-containing GalCer NT) equally well, but upon GTP hydrolysis, which promotes disassembly, the faster GTP hydrolyzing variants (and thus faster GTP reloading/cycling variants) appear to reassociate with membranes significantly better than the slower ones (Δ CT4/6>WT>CT+). Negative-stain EM data in Supplementary Fig. 7 further corroborate this. However, more sophisticated biophysical experiments beyond this study's scope are necessary to dissect the exact mechanisms at play quantitatively.

6. ***In Fig 3b, Δ CT4 and Δ CT6 mutants show almost comparable GTPase activity compared to WT but the fission probability and assembly properties appear much less in WT. Can the authors comment on the apparent disconnect between GTPase activity, assembly and fission.***

A major take-home point from our study is that there is *no correlation* between lipid-stimulated GTPase activity, assembly, and fission, which we now discuss at length in the Discussion section. Interestingly, this lack of correlation was raised previously by the Robinson group on the regulation of classical dynamin 1 (Dyn1) structure and activity by SH3 domain-containing binding partners¹⁷. This disconnect is further exemplified by an I533A mutation in the Dyn1 PH domain, which impairs the ordered self-assembly of Dyn1 on membranes *in vitro*¹⁸ and correspondingly fails in mediating endocytic vesicle scission *in vivo*¹⁹ yet manages to promote NT fission under favorable conditions *in vitro*^{19,20}. While we show an inverse correlation between the GTPase activity of various full-length Drp1 variants and fission (Supplementary Fig. 8), the Δ CT4 and Δ CT6 mutants, which on lipid NTs show markedly greater GTPase activity than WT are also ineffective in fission relative to the CT+ variant. This is likely because of the absent CT-SLiM-mediated ordered self-assembly interactions and progressive high-curvature generation required for fission. Consistent with this notion, the Δ CT variants form helical polymers of variable diameter on NTs even in the constant presence of GTP (Fig. 4c).

7. ***In Fig 4c, provide the statistics with a plot of number of tubes observed vs diameter of each tube in WT, CT+ and Δ CT4.***

Done. We have replotted the data for a better comprehension of our results.

8. ***Fig S9B, show a control of GIPC-1 + lipid.***

Done. We show it in Supplementary Fig. 10a (top right).

Minor changes:

9. ***Fig 1C. It would be nice to have a label helical diameter below the figure (first part of 1C).***

Done. Thanks for pointing it out.

10. ***Fig 2A. Please increase the font of the statistics and add the values in the text.***

Done. Thanks for the suggestion.

11. ***Fig 2B. Helix of CT+ can be labelled better so that it becomes easier to differentiate. An addition of overlay between CT+ and WT would be better to observe the inward buckling of G domain and the differences observed.***

Done. We have now moved the AlphaFold data to Supplementary Fig. 5.

References cited

1. Dar, S., Kamerkar, S.C. & Pucadyil, T.J. Use of the supported membrane tube assay system for real-time analysis of membrane fission reactions. *Nat Protoc* **12**, 390-400 (2017).
2. Martinez Galvez, J.M., Garcia-Hernando, M., Benito-Lopez, F., Basabe-Desmonts, L. & Shnyrova, A.V. Microfluidic chip with pillar arrays for controlled production and observation of lipid membrane nanotubes. *Lab Chip* **20**, 2748-2755 (2020).

3. Mahajan, M. et al. NMR identification of a conserved Drp1 cardiolipin-binding motif essential for stress-induced mitochondrial fission. *Proc Natl Acad Sci U S A* **118**(2021).
4. Hatch, A.L., Ji, W.K., Merrill, R.A., Strack, S. & Higgs, H.N. Actin filaments as dynamic reservoirs for Drp1 recruitment. *Mol Biol Cell* **27**, 3109-3121 (2016).
5. Macdonald, P.J. et al. A dimeric equilibrium intermediate nucleates Drp1 reassembly on mitochondrial membranes for fission. *Mol Biol Cell* **25**, 1905-15 (2014).
6. Ramonett, A. et al. Regulation of mitochondrial fission by GIPC-mediated Drp1 retrograde transport. *Mol Biol Cell* **33**, ar4 (2022).
7. Frohlich, C. et al. Structural insights into oligomerization and mitochondrial remodelling of dynamin 1-like protein. *EMBO J* **32**, 1280-92 (2013).
8. Lu, B. et al. Steric interference from intrinsically disordered regions controls dynamin-related protein 1 self-assembly during mitochondrial fission. *Sci Rep* **8**, 10879 (2018).
9. Rochon, K. et al. Structural Basis for Regulated Assembly of the Mitochondrial Fission GTPase Drp1. *bioRxiv*, 2023.06.22.546081 (2023).
10. Amacher, J.F., Brooks, L., Hampton, T.H. & Madden, D.R. Specificity in PDZ-peptide interaction networks: Computational analysis and review. *J Struct Biol X* **4**, 100022 (2020).
11. Perez-Jover, I., Madan Mohan, P., Ramachandran, R. & Shnyrova, A.V. Reply to Roy and Pucadyil: A gain of function by a GTPase-impaired Drp1. *Proc Natl Acad Sci U S A* **119**, e2202391119 (2022).
12. Kamerkar, S.C., Kraus, F., Sharpe, A.J., Pucadyil, T.J. & Ryan, M.T. Dynamin-related protein 1 has membrane constricting and severing abilities sufficient for mitochondrial and peroxisomal fission. *Nat Commun* **9**, 5239 (2018).
13. Montessuit, S. et al. Membrane remodeling induced by the dynamin-related protein Drp1 stimulates Bax oligomerization. *Cell* **142**, 889-901 (2010).
14. Basu, K. et al. Molecular mechanism of DRP1 assembly studied in vitro by cryo-electron microscopy. *PLoS One* **12**, e0179397 (2017).
15. Loson, O.C. et al. Crystal structure and functional analysis of MiD49, a receptor for the mitochondrial fission protein Drp1. *Protein Sci* **24**, 386-94 (2015).
16. Bustillo-Zabalbeitia, I. et al. Specific interaction with cardiolipin triggers functional activation of Dynamin-Related Protein 1. *PLoS One* **9**, e102738 (2014).
17. Krishnan, S., Collett, M. & Robinson, P.J. SH3 Domains Differentially Stimulate Distinct Dynamin I Assembly Modes and G Domain Activity. *PLoS One* **10**, e0144609 (2015).
18. Mehrotra, N., Nichols, J. & Ramachandran, R. Alternate pleckstrin homology domain orientations regulate dynamin-catalyzed membrane fission. *Mol Biol Cell* **25**, 879-90 (2014).
19. Ramachandran, R. et al. Membrane insertion of the pleckstrin homology domain variable loop 1 is critical for dynamin-catalyzed vesicle scission. *Mol Biol Cell* **20**, 4630-9 (2009).
20. Shnyrova, A.V. et al. Geometric catalysis of membrane fission driven by flexible dynamin rings. *Science* **339**, 1433-6 (2013).

Reviewers' Comments:

Reviewer #1:

Remarks to the Author:

The authors have addressed all comments and concerns. Congratulations on the great manuscript!

Reviewer #2:

Remarks to the Author:

The authors have responded to the suggested comments in a convincing way and also all the necessary changes have been incorporated in the revised version. I believe that the presented article will be a significant addition in the field and therefore I accept the manuscript in the current version.